# Scaling Wearable Foundation Models

**Girish Narayanswamy** [○,†,1,3*]**, Xin Liu** [○,†,1]**, Kumar Ayush** [1]**, Yuzhe Yang** [1]**, Xuhai Xu** [1]**,
Shun Liao** [1]**, Jake Garrison** [1]**, Shyam Tailor** [1]**, Jake Sunshine** [1,3]**, Yun Liu** [1]**, Tim Althoff** [1,3]**,
Shrikanth Narayanan** [1]**, Pushmeet Kohli** [2]**, Jiening Zhan** [1]**, Mark Malhotra** [1]**,
Shwetak Patel** [1,3]**, Samy Abdel-Ghaffar** [1]**, Daniel McDuff** [†,1]

[○]Co-first, [†]Corresponding, [1]Google Research, [2]Google DeepMind, [3]University of Washington
girishvn@uw.edu, {xliucs,dmcduff}@google.com

## Abstract

Wearable sensors have become ubiquitous thanks to a variety of health tracking features. The resulting continuous and longitudinal measurements from everyday life generate large volumes of data. However, making sense of these observations for scientific and actionable insights is non-trivial. Inspired by the empirical success of generative modeling, where large neural networks learn powerful representations from vast amounts of text, image, video, or audio data, we investigate the scaling properties of wearable sensor foundation models across compute, data, and model size. Using a dataset of up to 40 million hours of in-situ heart rate, heart rate variability, accelerometer, electrodermal activity, skin temperature, and altimeter per-minute data from over 165,000 people, we create LSM, a multimodal foundation model built on the largest wearable-signals dataset with the most extensive range of sensor modalities to date. Our results establish the scaling laws of LSM for tasks such as imputation, interpolation and extrapolation across both time and sensor modalities. Moreover, we highlight how LSM enables sample-efficient downstream learning for tasks including exercise and activity recognition.

## 1 Introduction

Wearable devices that monitor physiological and behavioral signals have become ubiquitous. Increasing evidence suggests that these devices can significantly contribute to promoting healthy behaviors (Ringeval et al., 2020), detecting diseases (Yang et al., 2022), and enhancing the design and implementation of treatments (Munos et al., 2016). These devices generate large volumes of continuous, longitudinal, and multimodal data. However such wearable time-series data can be difficult for consumers and experts to interpret. To this end, algorithms have been developed to translate time-series sensor data into human-readable representations, such as step counts and heart rates.

Historically, such algorithms for wearable sensors have relied on supervised, discriminative models designed to detect specific events or activities (Lubitz et al., 2022). This approach faces several significant limitations. First, the *limited volume and severe data imbalance* of labeled events results in large amounts of valuable *unlabeled* data being left unused. Second, supervised models are typically trained for *a single task* (e.g., classification), producing representations that may not generalize well to other tasks. Third, training data is often collected from *small study populations* (usually involving only tens or hundreds of participants), leading to a lack of diversity in the data.

Self-supervised learning (SSL) using generic pretext tasks (Noroozi et al., 2017; Caron et al., 2018; Yang et al., 2023) can yield versatile representations that are useful for a wide range of downstream applications. SSL allows for the use of a much larger proportion of available data without being restricted to labeled data regions (e.g., a limited number of subjects who self-report labels for exercises/activities). These advantages have motivated efforts to apply similar training strategies to build models from large volumes of unlabeled wearable data (Adaimi et al., 2024; Thapa et al., 2024; Yuan et al., 2024; Abbaspourazad et al., 2023) (see Table 1 for a summary).

Building on this, the empirical and theoretical success of scaling laws in neural models (Kaplan et al., 2020; Bahri et al., 2024) suggests that model performance improves predictably as compute, data,

---

[*] Work done during an internship at Google.

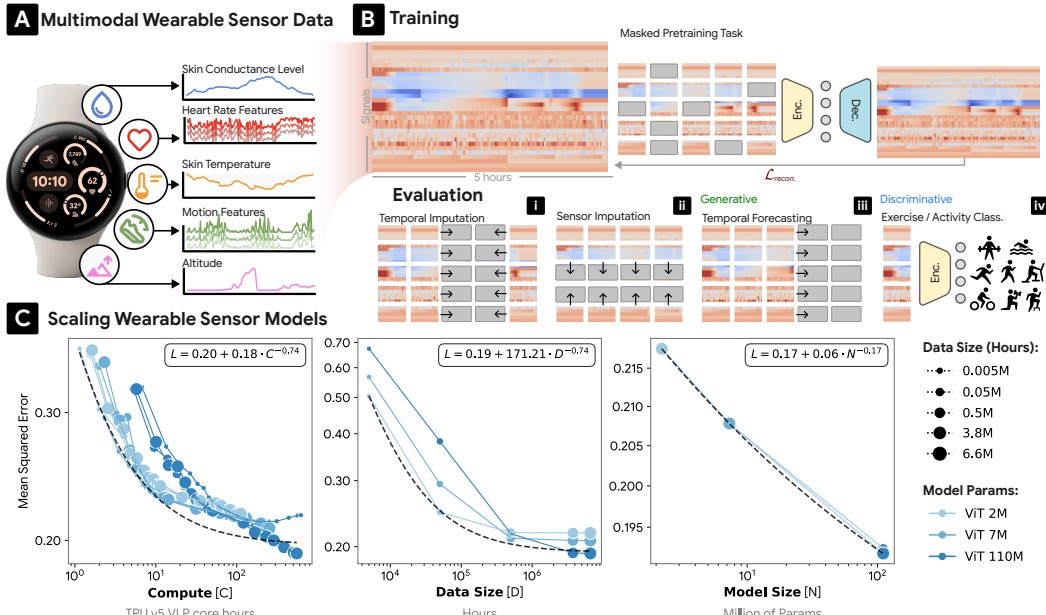

Figure 1: **Scaling Foundation Models on Wearable Data.** Self-supervised pretraining enhances the performance of models trained on wearable sensor data. **(A)** We present a systematic scaling analysis of sensor models using up to 40 million hours of multimodal data from over 165,000 people. **(B)** Using a random masking pretext task, we evaluate on tasks of imputation, forecasting, and downstream classification. **(C)** Experiments show scaling compute, data, and model size are all effective. Scaling, in this figure, is shown on the random imputation task.

and model parameters increase. These findings raise a critical research question: **Do scaling laws apply to models trained on wearable sensor data?** We aim to investigate whether the principles that drive the scaling of neural networks in domains like language and vision also extend to large-scale, multimodal wearable sensor data. Understanding how scaling manifests in this context could not only shape model design but also enhance generalization across diverse tasks and datasets.

In this paper, we present the results of our scaling experiments on the largest and the most diverse wearable dataset published to date, comprising up to 40 million hours of multimodal sensor data from over 165,000 users (Fig. 1). Leveraging this data, we train a foundation model, referred to as the **Large Sensor Model** (LSM), which is designed to capture generalizable representations across diverse populations, wearable sensor modalities, and downstream tasks. We demonstrate the scaling properties of LSM with respect to compute, data size, and model parameters, leading to substantial performance gains on generative imputation, interpolation, and extrapolation, as well as on downstream discriminative tasks. Our contributions can be summarized as follows:

- Implementation of the largest study to date on the scaling of sensor models, encompassing 40 millions hours, over 165,000 users, and multiple sensor modalities, including photoplethysmography (PPG), accelerometer, electrodermal activity (EDA), skin temperature, and altimeter signals.
- Identification of key strategies for training large-scale wearable sensor foundation models (LSM), and the scaling properties of LSMs with respect to compute, data size, and model parameters.
- Demonstration of LSM's ability to impute, interpolate, and extrapolate across temporal and sensor modalities, with a particular focus on generalization to unseen users.
- Verification that the LSM learned representations can be applied to downstream classification tasks, such as exercise and activity recognition, using ecologically valid, user-annotated events.

## 2 RELATED WORK

**Sensor Foundation Models.** Recent advances have demonstrated improved accuracy, robustness, and generalizability of models for sensor data by utilizing self-supervised pretraining on large-scale corpora of behavioral and physiological signals. Existing sensor foundation models primarily lever-

Table 1: **Comparisons of Studies on Wearable Sensor Foundation Models.**

| Study | #People (000s) | #Hours (000s) | ECG | PPG | ACC | SCL | TMP | ALT | Generative |
|---|---|---|---|---|---|---|---|---|---|
| Adaimi et al. (2024) | 0.05 | 0.20 | ✗ | ✗ | ✓ | ✗ | ✗ | ✗ | ✓ |
| Abbaspourazad et al. (2023) | 141 | 400 | ✓ | ✓ | ✗ | ✗ | ✗ | ✗ | ✗ |
| Yuan et al. (2024) | 100 | 15,700 | ✗ | ✗ | ✓ | ✗ | ✗ | ✗ | ✗ |
| LSM (**Ours**) | **165** | **40,000** | ✗ | ✓ | ✓ | ✓ | ✓ | ✓ | ✓ |

ECG: Electrocardiography, PPG: Photoplethysmography, ACC: Accelerometer,
SCL: Skin Conductance Level, TMP: Skin Temperature, ALT: Altimeter

age contrastive learning, creating positive and negative data pairs. Yuan et al. (2024) employ time domain augmentations (e.g., reversal, warping, permutation) to formulate the SSL task for motion data. Abbaspourazad et al. (2023) adopt a similar strategy, incorporating Gaussian noise, time and magnitude warping, and channel swapping. Thapa et al. (2024) generate data pairs using different sensory modalities. Most recently, RelCon (Xu et al., 2025) extends these ideas to show the utility of a relative contrastive loss in building a model more robust to false negatives and positives. In contrast, we focus on *masked input modeling* due to its generative capabilities and explore the resulting properties when scaling compute, data, and model size. Compared to prior work we consider more sensor inputs, a larger data sample, and systematically investigate scaling laws (Table 1).

**Time-Series Foundation Models.** Wearable sensor data typically takes the form of multivariate time-series signals. Historically, time-series models such as TiDE (Das et al., 2023), PatchTST (Nie et al., 2023), and TimesNet (Wu et al., 2023) have focused on tasks such as anomaly detection, and single target forecasting and imputation in specific domains such as energy use, transportation, finance, and climate. More recently, models such as TimesFM (Das et al., 2024), Chronos (Ansari et al., 2024), and MOMENT (Goswami et al., 2024) have shown the utility of self-supervised pre-training in building representations that better generalize to diverse applications. Recent works have also explored the potential of language foundation models to zero-shot reason on temporal data (Liu et al., 2023; Merrill et al., 2024), and to bootstrap time-series models (Zhou et al., 2023). Despite these advances, signals from disparate domains may exhibit considerably different properties. As such, we focus our exploration of generalist models in the *wearable sensor* domain.

**Scaling Laws in Deep Learning.** The scaling of computational resources, data volume, and model size has driven remarkable advances in deep learning (Zhai et al., 2022; Kaplan et al., 2020; Xie et al., 2023). Recent investigations indicate that testing loss follows a power law relationship with each of these three resources when the other two are held constant (Kaplan et al., 2020). Power law behavior has been observed across various domains, including large language models (Kaplan et al., 2020), large vision models (Zhai et al., 2022), transfer learning (Hestness et al., 2017), and multimodal models (Aghajanyan et al., 2023). In this work, we further this research direction and investigate the scaling behavior of models trained with multimodal *wearable sensor* data.

## 3 DATA FOR WEARABLE FOUNDATION MODELS

### 3.1 SENSOR DATA AND PROCESSING

Fitbit Sense 2 and Pixel Watch 2 have five *sensors* of highest relevance to this work: photoplethysmography, accelerometer, skin conductance, skin temperature, and altimeter/pressure sensors. From these input signals we compute a set of 26 *signals* (features), as described in Table 18 of Appendix G. Raw sensor data is not stored at this scale as it would impact the battery life and memory on the device. Thus, we focus on one-minute resolution signals (see Appendix G for additional discussions).

PPG **Photoplethysmography.** A validated algorithm (Nissen et al., 2022) is used to extract heart rate (HR) once per second via PPG. The per-minute HR data was calculated by taking the mean of

the interpolated, per-second data across non-overlapping one-minute windows. An on-device peak detection algorithm identified PPG-based R-wave peaks from which RR intervals were calculated. RR intervals are susceptible to noise from multiple sources, including movement, electronic noise, and missed heartbeats. To account for noise, outliers were removed using a sliding 5-minute window median-filter (Natarajan et al., 2020). The percentage of valid RR intervals for the 5-minute window is then calculated. Nine standard heart rate variability (HRV) metrics (Shaffer & Ginsberg, 2017) are calculated every minute over a sliding 5-minute window: RR $80^{th}$ percentile, RR $20^{th}$ percentile, RR median, RR mean, RR Shannon Entropy, RR differences Shannon Entropy, percentage of RR intervals greater than 30ms (PNN30), root mean squared difference of RR intervals, standard deviation of RR, and a boolean indicator of whether the optical sensor was on the wrist.

**ACC** **Accelerometer.** Ten signals are extracted from the 3-axis accelerometer: jerk, steps, accelerometer log energy, covariance, log energy ratio, the mean and standard deviation number of zero crossings, arm-tilt, kurtosis, and sleep coefficient. These signals are extracted by converting the 3-axis accelerometer to root mean squared magnitude (1D) and applying a high-pass filter (HPF) to the remove the DC component. In parallel, the 3-axis accelerometer signal is put through a second-order band-pass filter (BPF) and the principal component of the filtered 3-axis signal covariance matrix is calculated and updated every 25 seconds. Jerk is a measure based on the time-derivative of the acceleration calculated from the principal component. It is the logarithm of the ratio of the absolute of the t=1 autocorrelation lag over the t=0 autocorrelation lag. Steps is a per-minute count of steps taken calculated based on a machine learned classifier. Log energy is the logarithm of the sum of the squared HPF signal over the window. Covariance is the log of the condition number of the acceleration covariance matrix. Log energy ratio is the logarithm of the ratio of energy computed from principal-component over the magnitude of the HPF signal. Zero-crossings are the number of crossings in the principal component. Mean and standard deviation are calculated from the sample window. Arm tilt is the log of mean square root of squared X and Z axes. Kurtosis is calculated from the BPF signal. Each of these features is originally computed every 51 seconds and then resampled to a minutely resolution. Sleep coefficient is calculated as the sum of the 3-axes max-min range and binned into 16 log scaled bins before being input into a classifier to predict sleep probability.

**SCL** **Skin Conductance.** The electrodermal activity (EDA) sensor is used to infer sympathetic arousal via changes in micro-sweat levels, a physiological response to stress. Two electrodes on the back of the device measure changes in skin conductance level (SCL), which varies with skin moisture levels. SCL data is sampled at 200 Hz, downsampled to 25 Hz via a boxcar filter, and smoothed with a 5-minute median low-pass filter (McDuff et al., 2024). Per-minute tonic SCL slope and magnitude are then calculated. Due to the nature of the sensing mode operation, SCL data is only collected during non-exercise wake-periods.

**TMP** **Skin Temperature.** A temperature sensor located near the wrist-facing surface of the device takes measurement every 10 seconds. Per-minute slope and magnitude values are calculated via linear regression. Skin temperature signals are available whenever EDA signals are available.

**ALT** **Altimeter.** The standard deviation of the altimeter (pressure sensor) measurements.

All sensor signals were globally normalized (z-score) to remove differences in magnitude due to different units of measurement. As the masked autoencoder cannot process missing data, we imputed minutes that had missing values. Within each 300-minute window, missing data between valid data points was linearly interpolated, and leading missing minutes were backfilled.

## 3.2 BUILDING A LARGE SCALE PRETRAINING SENSOR DATASET

To build the large dataset for our experiments we sampled wearable data from 165,090 subjects during the period January $1^{st}$ 2023 to July $2^{nd}$ 2024. The subjects wore Fitbit Sense 2 or Google Pixel Watch 2 devices and consented for their data to be used for research and development of new health and wellness products and services. We sub-selected individuals wearing one of these devices as older device generations included fewer sensors. The subjects were asked for self-reported sex, age and weight. Table 2(a) summarizes the characteristics of the pretraining data. All data were de-identified and not linked with any other information. To create a dataset that maximized the number of subjects we randomly sampled 10 5-hour windows of data from each subject, for a total of 8 million hours (6.6 million pretrain hours, 1.7 million test hours). We explore the extremes of data scaling by experimenting with a subject-imbalanced 40 million hour pretraining dataset

Table 2: **Details of the Datasets.** Summary of the demographic composition of our pretraining set and class distribution of our downstream set samples.

(a) **Demographics of the Pretraining Set.**

| | Category | # People | % |
|---|---|---|---|
| **Sex** | Female | 110,780 | 67.1% |
| | Male | 53,895 | 32.6% |
| | Not Specified | 415 | 0.3% |
| **Age** | 18-39 | 55,653 | 33.7% |
| | 40-59 | 75,627 | 45.8% |
| | 60-79 | 32,251 | 19.6% |
| | $\geq$80 | 1,548 | 0.9% |
| | Not Specified | 11 | 0.0% |
| **BMI** | Healthy ($<$25) | 57,015 | 34.6% |
| | Overweight (25-30) | 52,950 | 32.1% |
| | Obese ($\geq$30) | 54,727 | 33.1% |
| | Not Specified | 398 | 0.2% |
| | **Total** | 165,090 | 100% |

(b) **Class Distribution of the Downstream Set.**

| Class | # Training | # Testing |
|---|---|---|
| Exercise | 3,272 | 671 |
| Non-Exercise | 6,195 | 1,329 |
| **Total** | 9,467 | 2,000 |
| Biking | 1,191 | 412 |
| Elliptical | 152 | 49 |
| High Intensity Training | 332 | 104 |
| Strength Training | 229 | 425 |
| Swimming | 2,332 | 441 |
| Running | 1,860 | 315 |
| Walking | 7,607 | 1,418 |
| Weightlifting | 669 | 98 |
| **Total** | 14,372 | 3,262 |

(Appendix C.1). Note that these datasets are comprised of wearable data from daily-living, including diverse timestamps and a range of life events, and are not biased toward specific events or activities.

The dataset was split 80-20, based on subjects, into train-test splits (132072 subjects in training, 33018 subjects in testing) as described in Table 2(a). We then created several "slices" of the training set to conduct the scaling experiments. The test set remains identical throughout all experiments. In the "sample-scaling" experiments we shuffled the training data and took $N$ samples per experiment. In the "subject-scaling" experiments we grouped the training data by subject identifier and took all samples from $M$ subjects per experiment.

## 4 SENSOR MODELING TASKS

### 4.1 GENERATIVE TASKS

We posit that defining generative tasks in the training of wearable sensor models may not only result in learned representations that are useful for downstream classification tasks, but also produce models that can impute missing or incomplete data (interpolate) and extrapolate future sensor values (forecast). To train the model and to test these capabilities we define several tasks (see Fig. 2).

**Random Imputation.** Our primary pretext task involves removing patches randomly from the input sample across the time-axis and signal-axis. During training this requires the model to infer missing values and make predictions based on the partial input.

**Temporal Interpolation.** Sensor inputs can be missing for a number of reasons. Devices need to be removed from the wrist for charging, and certain sensors might be turned off for periods to save on battery life (McDuff et al., 2024). Interpolation of sensor data is an important and necessary step for many algorithms. In this task we test the model's ability to fill gaps in the data where all sensor data is missing for a period of time, usually between two observations.

**Sensor Imputation.** Sensor imputation refers to the process of inferring a subset of partially missing sensor-streams, from other continuously online sensing modalities. By leveraging correlations between different physiological signals, sensor imputation ensures that insights can be derived even when some sensor modalities are absent, enhancing the overall versatility and capabilities of multi-sensor systems. Under the constraints of hardware limitations (battery, wireless connectivity, etc.), sensor imputation can enable the delivery of more realistic metrics to the user (e.g., step count, average resting heart rate) even if when sensors are not continuously online.

**Temporal Extrapolation (Forecasting).** A more challenging task than interpolation is extrapolation of sensor values forward in time. Temporal extrapolation involves predicting future sensor

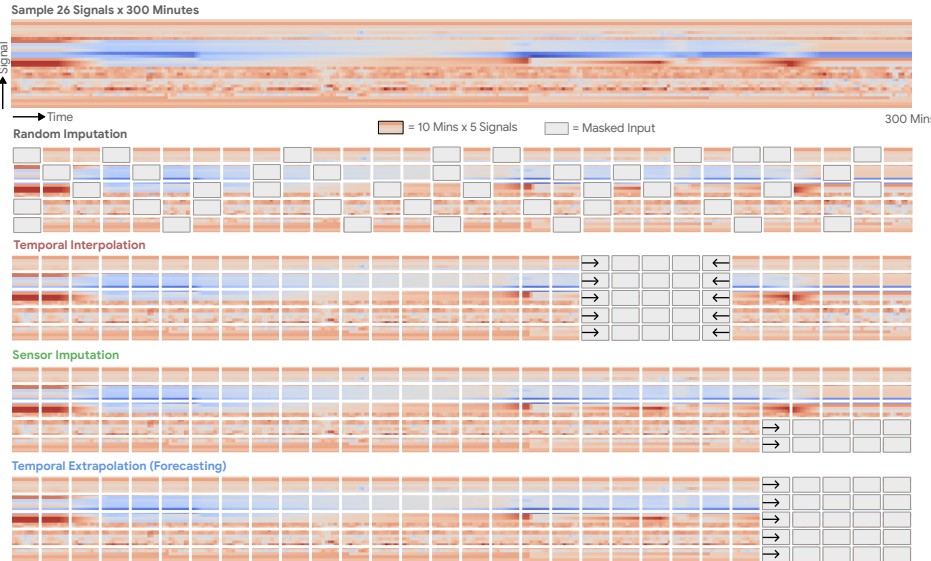

Figure 2: **Generative** LSM **Tasks and Pretraining.** We define four distinct generative tasks: random imputation, temporal interpolation, signal/sensor imputation, and temporal extrapolation (forecasting). Random imputation was empirically chosen as the pretraining task.

measurements. The ability to anticipate future physiological states based on current and historical data has applications in areas such as health interventions, where extrapolation can be used to schedule recovery times, detect early signs of fatigue, predict wake-up times, and detect anomalies. Accurate signal extrapolation is a key task that can empower wearable devices to provide more just-in-time, proactive, and personalized health recommendations.

## 4.2 DISCRIMINATIVE TASKS

Discriminative tasks focus on classifying or identifying specific activities, states, or conditions based on sensor data. These tasks are essential for translating raw sensor inputs into actionable, personalized, and relevant feedback. The corresponding datasets are split 80-20 into user-stratified train-test splits unless otherwise specified. Two exemplary tasks are considered here. Additional discriminative tasks (sex, binned age, and subject dependent mood) and results can be found in Appendix C.7.

**Activity Recognition.** Activity recognition is the process of classifying different user activities such as biking, running, or walking, based on the patterns detected in sensor data. This allows wearable devices to monitor daily routines accurately, providing insights into fitness levels, activity trends, and overall health. Effective activity recognition enables applications like fitness tracking, lifestyle monitoring, and personalized coaching. Our dataset includes eight user-labeled activities: *Biking*, *Elliptical*, *High-Intensity Interval Training (HIIT)*, *Strength Training*, *Swimming*, *Running*, *Walking*, and *Weightlifting*. The activity event labels are self-reported by users post hoc.

**Exercise Detection.** Exercise detection identifies when a user is exercising, enabling real-time feedback and performance tracking. This task involves recognizing exercise events from continuous sensor data, allowing devices to log workout sessions, track progress, and provide personalized recommendations. Additionally, detecting exercise unlocks related experiences, such as identifying exercise types, marking session start times, or tracking post-exercise feedback. We developed a dataset with windows of user-labeled exercise and non-exercise events (see Table 2(b)). The exercise event labels are based on self-reported activity labels.

## 5 EXPERIMENTS & RESULTS

### 5.1 TRAINING PROCEDURES

We pretrain our wearable foundation models on a diverse collection of multimodal sensor data. Each sample is processed as a two-dimensional matrix of 26 signals by 300 minutes (see Fig. 2). Our pri-

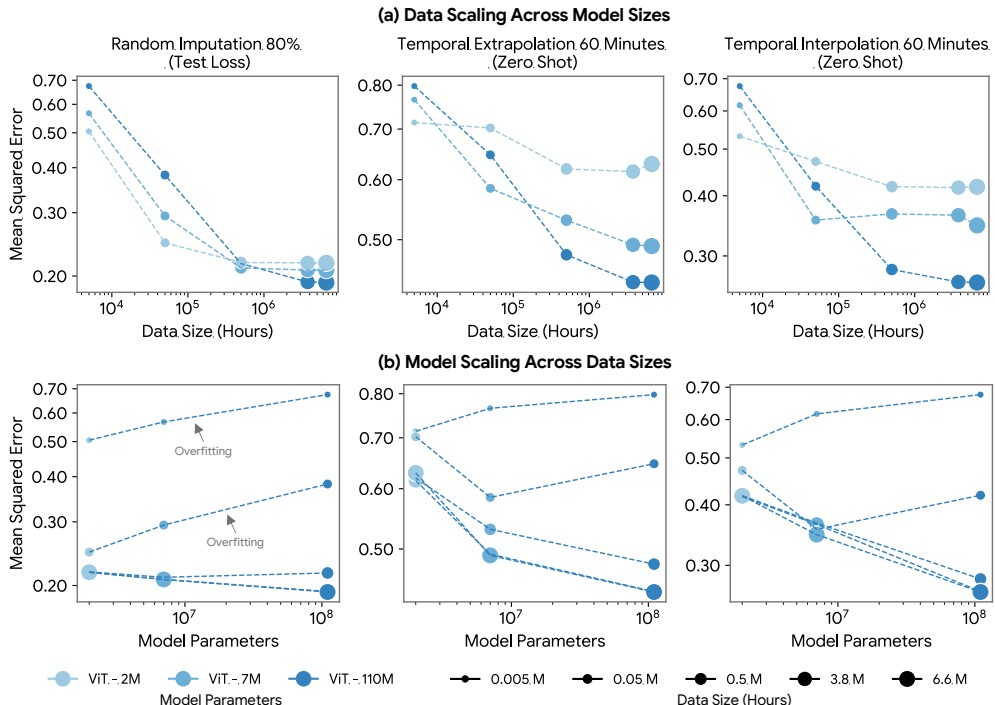

Figure 3: **Scaling Performance of** LSM**.** We show performance on *generative tasks* across varying data and model sizes. LSM begins to saturate at approximately $10^7$ hours of data. The effects of scaling are more pronounced in imputation, interpolation, and extrapolation tasks. Results indicate that as model size increases, significantly larger data volumes are required to prevent overfitting.

mary pretraining objective is to optimize the masked signal reconstruction loss (i.e., mean squared error), averaged over randomly masked patches from the input matrix (He et al., 2022). The primary performance metric is the mean squared error on the held-out test set, evaluated across all the normalized signals. We employ a masked autoencoder (MAE) due to its ability to handle multitarget generative tasks like forecasting and interpolation, and discriminative tasks such as activity recognition, unlike traditional time-series models that focus on single-target prediction (Ansari et al., 2024). Furthermore, MAE gracefully handles missingness which is inherent in wearable data. MAE has proven effective in pretraining and scalable learning across domains (Huang et al., 2022; Xie et al., 2023), which aligns with our focus on establishing scaling laws in the wearable data domain.

We pretrain our models on Google v5e TPUs with a total batch size of 4,096 across 50,000 training steps. The training process uses the AdamW optimizer with a base learning rate of $5e-3$ and weight decay set to $1e-4$. A linear warm-up schedule is applied for the first 2,500 steps, followed by a cosine learning rate decay to zero. All pretraining experiments use an 0.8 masking ratio (masking out random patches that cover 80% of the total input signals). Additional details on implementation and hyperparameters can be found in Appendix E.

## 5.2 RESULTS & DISCUSSION

**Do Scaling Laws Apply to Wearable Data?** We present the Pareto front of the downstream reconstruction loss as a function of resource scaling (compute, data size, model size) (Fig. 1(c)). Over multiple orders of magnitude, the relationship between the scaled resource and performance follows a power-law ($L = aC^b$), resulting in a near-linear trend on the log-log plot. Similar to behavior observed for scaling language (Henighan et al., 2020) and vision models (Zhai et al., 2022), we observe saturation, where the loss does not asymptotically approaches zero; we therefore include an additive constant $c$ to model this effect: $L = aC^b + c$.

We highlight *compute scaling* across data and model sizes in Fig. 1(c), and find that though effective, even the largest model, trained with the largest data volume, experiences some saturation effect.

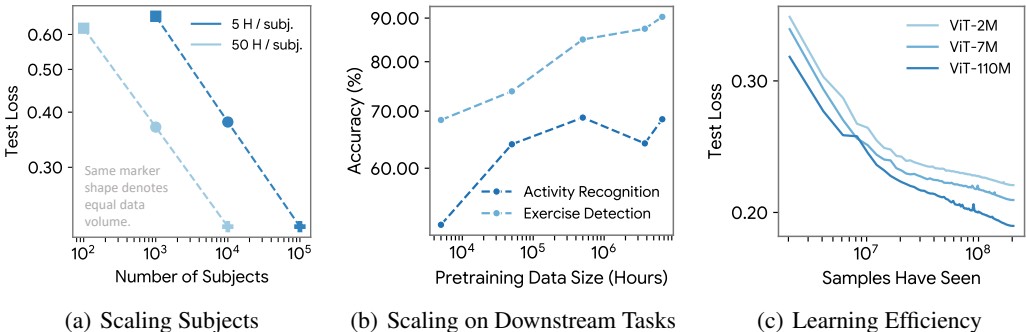

(a) Scaling Subjects       (b) Scaling on Downstream Tasks       (c) Learning Efficiency

Figure 4: **Analysis on Scaling** `LSM`. **(a)** Performance improves with both scaling subject count and data hours, with longer durations yielding additional gains (additional insights in Section 5.2). **(b)** Data scaling on discriminative tasks with ViT-110M. **(c)** Larger models are more sample efficient.

We illustrate *data scaling* across various model sizes (Fig. 3(a)). Performance improves monotonically to approximately $10^5$ data hours, beyond which the rate of improvement diminishes, particularly around $10^7$ hours. We validated that scaling beyond $10^7$ hours yields minimal benefits by training with 40 million hours (see Appendix C.1). Consequently, results in Table 3 are pretrained with 6.6 million hours of data. Larger models, especially the ViT-110M, continue to benefit from data scaling, showing substantial gains when training on over 1 million hours of data. These observations underscore the large data requirements needed to fully exploit the capacity of larger models, which are far greater than those required by smaller models. A similar trend is observed in discriminative tasks (Fig. 4(b)). We further note that these trends are based on minutely aggregated wearable data; raw sensor signals are traditionally collected at substantially higher sampling frequencies and it is possible that feature extraction on more fine-grained sensor data may require even larger models.

*Model scaling* results as a function of data size demonstrate that as both model size and dataset size are scaled, sufficient data is essential to prevent overfitting (Fig. 3(b)). Models trained on smaller datasets exhibit limited generalization capacity, whereas scaling up to $10^8$ parameters results in significant gains in test loss and generative zero-shot performance. These findings highlight the need to align model size with adequate data to fully leverage the model's representational power. Our experiments also show larger models are more sample efficient as illustrated in Fig. 4(c).

By scaling compute, data, and model size together, `LSM` achieves improvements of 16% to 23% in temporal interpolation MAE and 20% to 21% in extrapolation MAE across five time durations as compared to the best baseline method (Table 3(a)). Additionally, `LSM` outperforms baselines in exercise detection and 8-class activity recognition improving upon the supervised baseline by 27% and 29% in accuracy and 57% and 54% in mAP, as detailed in Table 3(c). Such baseline approaches are commonly used by existing sensor algorithms (Gershon et al., 2016; van Rossum et al., 2023). More scaling results can be found in Appendix C.

**Is Scaling Subjects or Wearable Data Hours per Subject More Helpful?** Fig. 4(a) demonstrates that both scaling strategies are equally important, though a minor benefit is observed using additional per-subject data, especially when the total data-volume is low. One possible hypothesis to explain this effect is that the diversity of activities per-subject (as reflected by increased hours per-subject) plays a crucial role. Alternatively, inter-subject diversity may become more important when learning representations for longer context windows (e.g., days or weeks as opposed to hours). While additional subject-hours are crucial for capturing intra-subject variability, increasing the number of subjects introduces valuable inter-subject diversity. Therefore, scaling both dimensions, subjects and hours, are essential and valid strategies to fully leverage the model's capacity and realize gains.

**Can Wearable Foundation Models Impute the Past and Predict the Future?** As shown in Fig. 3, scaling laws apply to imputation, interpolation, and extrapolation tasks, with larger models and more data resulting in improved performance. The utility of `LSM` is further emphasized in Table 3(a). However, despite these quantitative gains, the qualitative results in Fig. 11 of Appendix C.6 reveal that these tasks remain highly challenging. Imputing large portions of missing data, especially over extended time intervals, often leads to degraded accuracy, with performance deteriorating as the missing data window increases. Similarly, extrapolation further into the future (e.g., several hours

Table 3: **Comparisons of** LSM **and Baseline Methods on Generative and Discriminative Tasks.**

(a) **Generative Task Results.**

| Task + Method | Error (MAE / MSE) | | | | |
|---|---|---|---|---|---|
| **Temporal Interpolation** | **10 mins** | **20 mins** | **30 mins** | **60 mins** | **120 mins** |
| MEAN | 0.36 / 0.42 | 0.36 / 0.43 | 0.37 / 0.44 | 0.38 / 0.46 | 0.39 / 0.49 |
| MICE | 0.36 / 0.42 | 0.36 / 0.43 | 0.37 / 0.43 | 0.38 / 0.45 | 0.39 / 0.48 |
| NEAREST NEIGHBOR | 0.21 / 0.29 | 0.26 / 0.37 | 0.28 / 0.42 | 0.33 / 0.51 | 0.38 / 0.62 |
| LINEAR INTERP. | 0.19 / 0.23 | 0.23 / 0.30 | 0.26 / 0.34 | 0.30 / 0.42 | 0.36 / 0.51 |
| LSM (MAE) | **0.16 / 0.14** | **0.19 / 0.18** | **0.20 / 0.21** | **0.24 / 0.26** | **0.29 / 0.33** |
| GAINS OVER INTERP. | **+16% / 39%** | **+17% / 40%** | **+23% / 38%** | **+20% / 38%** | **+19% / 33%** |
| **Temporal Extrapolation** | **10 mins** | **20 mins** | **30 mins** | **60 mins** | **120 mins** |
| MEAN | 0.48 / 0.66 | 0.48 / 0.65 | 0.47 / 0.65 | 0.47 / 0.64 | 0.45 / 0.64 |
| MICE | 0.48 / 0.66 | 0.48 / 0.66 | 0.47 / 0.65 | 0.47 / 0.65 | 0.45 / 0.64 |
| NEAREST NEIGHBOR | 0.35 / 0.52 | 0.40 / 0.62 | 0.43 / 0.68 | 0.47 / 0.76 | 0.48 / 0.81 |
| LINEAR INTERP. | 0.35 / 0.52 | 0.40 / 0.62 | 0.43 / 0.68 | 0.47 / 0.76 | 0.48 / 0.81 |
| LSM (MAE) | **0.28 / 0.31** | **0.32 / 0.37** | **0.34 / 0.40** | **0.37 / 0.44** | **0.38 / 0.47** |
| GAINS OVER INTERP. | **+20% / 40%** | **+20% / 40%** | **+21% / 23%** | **+21% / 31%** | **+21% / 27%** |
| **Sensor Imputation** | **10 mins** | **20 mins** | **30 mins** | **60 mins** | **120 mins** |
| MEAN | 0.36 / 0.42 | 0.36 / 0.43 | 0.37 / 0.43 | 0.38 / 0.45 | 0.39 / 0.49 |
| MICE | 0.30 / 0.33 | 0.30 / 0.36 | 0.31 / 0.38 | 0.33 / 0.46 | 0.37 / 0.61 |
| NEAREST NEIGHBOR | 0.21 / 0.29 | 0.26 / 0.37 | 0.28 / 0.42 | 0.33 / 0.51 | 0.38 / 0.62 |
| LINEAR INTERP. | 0.19 / 0.23 | 0.23 / 0.30 | 0.26 / 0.34 | 0.30 / 0.42 | 0.36 / 0.51 |
| LSM (MAE) | **0.15 / 0.11** | **0.15 / 0.12** | **0.16 / 0.13** | **0.17 / 0.15** | **0.19 / 0.17** |
| GAINS OVER INTERP. | **+21% / 52%** | **+35% / 60%** | **+38% / 62%** | **+43% / 64%** | **+47% / 67%** |

(b) **Exercise Detection.**

| Pretrain | Probe/FT | Acc. | mAP |
|---|---|---|---|
| - | RAND. FOREST | 73.0 | 76.8 |
| - | LOGISTIC REG. | 72.4 | 67.3 |
| - | SUPERVISED | 70.9 | 61.7 |
| MSN | Linear Probe | 67.6 | 60.0 |
| DINO | Linear Probe | 66.0 | 57.0 |
| SIMCLR | Linear Probe | 66.5 | 51.5 |
| LSM (MAE) | Linear Probe | 84.7 | 89.0 |
| MSN | Fine-tune | 76.7 | 74.6 |
| DINO | Fine-tune | 78.2 | 80.3 |
| SIMCLR | Fine-tune | 74.9 | 66.6 |
| LSM (MAE) | Fine-tune | **90.3** | **97.0** |
| GAIN OVER SUPERVISED | | **+27%** | **+57%** |

(c) **Activity Recognition.**

| Pretrain | Probe/FT | Acc. | mAP |
|---|---|---|---|
| - | RAND. FOREST | 56.5 | 43.1 |
| - | LOGISTIC REG. | 60.4 | 39.3 |
| - | SUPERVISED | 53.2 | 33.4 |
| MSN | Linear Probe | 44.6 | 24.0 |
| DINO | Linear Probe | 50.3 | 26.0 |
| SIMCLR | Linear Probe | 45.3 | 20.8 |
| LSM (MAE) | Linear Probe | 49.4 | 24.6 |
| MSN | Fine-tune | 62.5 | 43.4 |
| DINO | Fine-tune | 66.2 | 46.3 |
| SIMCLR | Fine-tune | 67.3 | 46.0 |
| LSM (MAE) | Fine-tune | **68.5** | **51.4** |
| GAIN OVER SUPERVISED | | **+29%** | **+54%** |

All neural methods, including the supervised method, utilize a ViT-Base (110M) backbone. Relevant methods are pretrained with 6.6 million hours of data. In the sensor imputation task, we randomly mask 67% of the sensor modalities. MICE (Van Buuren & Groothuis-Oudshoorn, 2011), Random Forest (Breiman, 2001), MSN (Assran et al., 2022), DINO (Caron et al., 2021), SimCLR (Chen et al., 2020), MAE (He et al., 2022).

ahead) introduces significant uncertainty, making it difficult to predict fine-grained physiological or behavioral patterns. These findings suggest that while scaling helps improve generative capabilities, substantial challenges remain, particularly in handling long-range dependencies and large data gaps.

**Are Wearable Foundation Models Label Efficient on Discriminative Tasks?** Our experiments on probing, fine-tuning, and few-shot learning for activity indicate that wearable foundation models are highly label efficient. As shown in Table 3(b), 3(c), and Appendix Table 14, the performance of the fine-tuned LSM consistently outperforms supervised baselines. A confusion matrix of the best performing model is shown in Fig. 7 of the Appendix. As shown in Table 11 of Appendix C.2, even in the low-data regime (e.g., 5-shot, 10-shot), these models demonstrate strong generalization capabilities, achieving significantly lower error rates compared to models trained from scratch or with limited supervision. As the number of labeled examples increases, the performance gap widens, with foundation models leveraging pretraining to more effectively transfer learned representations

to downstream tasks. t-distributed Stochastic Neighbor Embeddings (t-SNE) plots illustrating the impact of pretraining on more data and fine-tuning are shown in Appendix C.4 (Fig. 8).

**Why did Models Saturate?** Our experiments indicate promising opportunities in scaling wearable sensor models but also highlight several unresolved questions. Notably, we observe saturation in scaling laws with a dataset size of $10^7$ hours and model sizes near 100M parameters. We attribute this to three factors: (1) The current pretraining task may not be sufficiently scalable. For example, it is possible that vision-domain methods are ill-suited to fully leverage sensor data, or that a decoder-only approach may outperform reconstructing masked inputs. (2) The dataset construction lacks sufficient challenge, and extending the sensor context window from 5 hours to a day or even a week could introduce more complexity that enables the model to learn longer time dependency relationships. (3) Our data cleaning process was minimal, and increasing data diversity, akin to large-scale language model training, could significantly enhance model generalization. For example, while our dataset spanned all four seasons, there was an imbalance in temporal coverage, with two years of data from January to June but only a single year from July to December. This uneven distribution could bias the model towards activities more common in the earlier part of the year.

**Additional Information.** Additional results of model design choices are presented in Appendix A. Additional related work is highlighted in Appendix B. Qualitative examples of signal and embedding are provided in Appendix C.4 & C.6. Discussions regarding scaling LSM to the edge, contextualizing large sensor models, and the utility of physiological proxy tasks can be found in Appendix D.

# 6 LIMITATIONS & FUTURE WORK

A key characteristic of wearable sensor data is its inherent missingness. Handling missing data in both pretraining and downstream tasks remains an open question. While we used imputation for this study, a more principled approach would involve designing models that naturally account for missing data without introducing imputation biases. The nature of missing data in wearable sensors often correlates with real-world events (e.g., charging the device, loose fitting), which can mean that data is missing not at random (MNAR). We recognize our study does deeply investigate the various types of missingness in wearable data; understanding these factors and designing methods to handle them robustly remains an important direction for future work. Lastly, we acknowledge the lack of comprehensive evaluation on more discriminative tasks and diverse sets of wearable devices. Future work will expand the dataset to include a broader range of classification and regression tasks across different dataset sources and devices.

We acknowledge that creating generic input representations for data from different sensors, each with unique properties, is challenging. This motivated our choice to build the model on minutely features derived from raw sensor data, which inherently vary in sampling rates. These features can be similarly derived from a wide range of sensors, making the inputs somewhat device-agnostic. However, wearable signals in their raw form (e.g., 120Hz accelerometer values) often exhibit strong periodicity, which is not retained in the aggregated minutely features. As a result, we focused on time-domain representations for our inputs, but we acknowledge that this approach may not fully leverage the fine-grained temporal or spectral characteristics of raw signals, potentially limiting generalization across diverse modalities and resolutions. We acknowledge that the 5-hour window represents a trade-off, balancing the complexity needed for pretraining with the temporal specificity required for downstream tasks. Future work will explore task-specific window sizes, refined masking strategies to better align with the temporal characteristics of downstream events, and the ability of these models to generalize across wearable devices with varying sensor capabilities.

# 7 CONCLUSION

We present LSM, a large multimodal foundation model trained on up to 40 million hours of wearable sensor data from over 165,000 individuals, establishing scaling laws for sensor models. LSM improves performance across generative tasks and discriminative tasks. Our results demonstrate that scaling data, model size, and compute leads to substantial gains in generalization and efficiency. LSM highlights the potential of scaling wearable sensor models for real-world health applications.

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

# APPENDIX

## A  MODEL DESIGN CHOICES AND ABLATIONS

We perform ablations on the configurations used for our masked autoencoder LSM design. Following the convention of previous works (He et al., 2022; Huang et al., 2022), we explore masking ratio, masking strategies, patch sizes, and model sizes. Uniquely, we explore the ordering of sensor signals, as these signals do not share the same explicit ordered dependencies as exist in images and audio spectrograms. We employ random masking, a 0.8 masking ratio, ordered sensor signal order, a patch size of 10x5, and a LSM-Base (110M) backbone, unless otherwise specified.

### A.1  SELECTING A MASKING RATIO

Selecting the appropriate masking ratio is critical for ensuring effective representation learning in our sensor MAE training. We explore different masking ratios, ranging from 30% to 90%, to evaluate their impact on reconstruction quality and model generalization. We find that a masking ratio of 80% yields the best performance on temporal interpolation and extrapolation as shown in Table 4.

Table 4: **Ablation Study of Masking Ratios.**

| Mask Ratio | Interpolation 60 mins | | Extrapolation 60 mins | |
|---|---|---|---|---|
| | MAE | MSE | MAE | MSE |
| 0.3 | 0.29 | 0.31 | 0.38 | 0.47 |
| 0.4 | 0.35 | 0.39 | 0.38 | 0.45 |
| 0.5 | 0.25 | 0.27 | 0.39 | 0.46 |
| 0.6 | 0.44 | 0.57 | 0.38 | 0.45 |
| 0.7 | 0.40 | 0.51 | **0.37** | **0.44** |
| 0.8 | **0.24** | **0.26** | **0.37** | **0.44** |
| 0.9 | 0.31 | 0.33 | 0.40 | 0.49 |

### A.2  SELECTING A MASKING STRATEGY

To train a wearable foundation model effective for both generative and discriminative tasks, mask-based pretraining proves superior to contrastive pretraining. Choosing the right masking strategy is crucial, as it directly influences the quality of the learned embeddings and the model's generalizability. In Table 5 we systematically compare five masking strategies and demonstrate that random masking yields the best performance across the two generative tasks. We find that structured temporal masking performs equivalently, and we select random masking to match prior work (He et al., 2022; Huang et al., 2022). Example visualizations of these masking strategies can be seen in Fig. 5.

Table 5: **Ablation Study of Masking Strategies.**

| Mask Strategy | Interpolation 60 mins | | Extrapolation 60 mins | |
|---|---|---|---|---|
| | MAE | MSE | MAE | MSE |
| RANDOM | **0.24** | **0.26** | **0.37** | **0.44** |
| STRUCTURED (TEMPORAL) | **0.24** | **0.26** | **0.37** | **0.44** |
| STRUCTURED (SENSOR) | 0.54 | 0.71 | 0.53 | 0.73 |
| TEMPORAL INTERPOLATION | 0.41 | 0.48 | 0.52 | 0.66 |
| TEMPORAL EXTRAPOLATION | 0.43 | 0.51 | 0.51 | 0.64 |

### A.3  SELECTING A PATCH SIZE.

Patch size, for our modeling, is defined by time steps and the number of sensor features per patch. In contrast to previous works (He et al., 2022; Huang et al., 2022) we sweep across both feature and

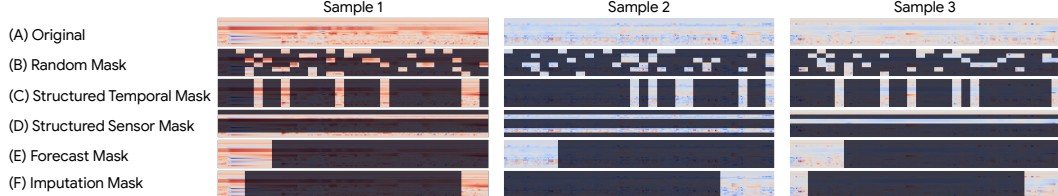

Figure 5: `LSM` **MAE Pretrain Masking Strategies.** All strategies employ a masking ratio of 0.8. **(A)** original, unmasked sensor matrix, **(B)** random masking, **(C)** structured temporal masking, **(D)** structured sensor masking, **(E)** temporal extrapolation masking, **(F)** temporal interpolation masking. Both random and structured temporal masking enable strong down-stream performance. We select random masking for all scaling experiments and evaluations.

temporal dimensions of the input. This is critical for sensor modeling, where both dimensions, of time and sensor signals, share unique correlations and dependencies along their corresponding axes.

A time-step of 10 minutes strikes the best balance of low gFlops (15.94) and strong performance (MAE of 0.24 for imputation, 0.37 for forecasting) (See Table 6). Similarly, five features per patch achieves the best trade-off between accuracy and computational cost, outperforming both smaller (10x1) and larger patches (10x26). Thus, a moderate patch size of 10 minutes by 5 features is selected. As a patch-size of 10-minutes x 5-sensors (10x5) cannot evenly patch an input sensor matrix of 300-minutes x 26-sensors (300x26), we zero-pad the sensor dimension to 30 resulting in an 300-minute x 30-feature (300x30) input sensor matrix.

We concede that 10x5 patching seems less grounded than 10x1 or 10x26. It is possible that for such small input windows (e.g., 300 minutes) local feature clustering is extremely important. It is further possible that this dependency may not exist for larger context windows (e.g., a day or a week).

Table 6: **Ablation Study of Patch Sizes.**

(a) **Sweep Across Time-Steps per Patch.** (5 feats. per patch)

| Patch Size | gFlops | Interpolation 60 mins | | Extrapolation 60 mins | |
|---|---|---|---|---|---|
| | | MAE | MSE | MAE | MSE |
| 5x5 | 33.09 | 0.34 | 0.41 | **0.37** | 0.46 |
| 10x5 | 15.94 | **0.24** | **0.26** | **0.37** | **0.44** |
| 20x5 | 7.82 | 0.26 | 0.28 | 0.41 | 0.48 |
| 30x5 | **5.18** | 0.28 | 0.30 | **0.37** | **0.44** |

(b) **Sweep Across Features per Patch.** (10 mins. per patch)

| Patch Size | gFlops | Interpolation 60 mins | | Extrapolation 60 mins | |
|---|---|---|---|---|---|
| | | MAE | MSE | MAE | MSE |
| 10x1 | 77.83 | 0.30 | 0.33 | 0.43 | 0.53 |
| 10x2 | 36.07 | 0.30 | 0.33 | 0.45 | 0.53 |
| 10x5 | 15.94 | **0.24** | **0.26** | **0.37** | **0.44** |
| 10x10 | 7.82 | 0.33 | 0.38 | 0.45 | 0.55 |
| 10x26 | **2.58** | 0.28 | 0.31 | 0.43 | 0.51 |

## A.4    SELECTING A SENSOR SIGNAL ORDER

For multimodal sensor data, the order in which signals are processed by the model can impact performance. Specifically, for architectures, such as vision transformers, that take patched inputs, the clustering of signals in patches may have a profound impact on the learned representation. We evaluate ordering the sensor signals by: (a) sensor types (as in Table 18), (b) randomized order (repeated with several random seeds) and (c) interleaving signals with uncorrelated signals. Cross correlation matrices are shown in Fig. 10. We find that ordering by clustering sensor type generally yields better results (see Table 7), particularly when dealing with heterogeneous sensor modalities like PPG, accelerometry, and EDA. We posit that this ordering allows the model to leverage specific sensor characteristics more effectively, improving performance on downstream tasks.

## A.5    MODEL SIZE VARIANTS

In Table 8 we present four variants of the `LSM` models we trained. The model sizes and naming conventions partially follow the tradition established by T5 (Raffel et al., 2020). Our results indicate that scaling the model beyond `LSM`-B offers no additional improvements in either reconstruction loss or downstream task performance. Based on this insight all neural methods in Table 3 employ a ViT-110M backbone.

Table 7: **Ablation Study of Sensor Orders.**

| Sensor Order | Interpolation 60 mins | | Extrapolation 60 mins | |
|---|---|---|---|---|
| | MAE | MSE | MAE | MSE |
| CLUSTERED | **0.24** | **0.26** | **0.37** | **0.44** |
| RANDOMIZED (N=5) | 0.28 | 0.32 | 0.38 | 0.45 |
| MAX ENTROPY | 0.30 | 0.34 | 0.45 | 0.55 |

Table 8: **Vision Transformer Variants used in** `LSM`. An LSM-[size] model indicates a ViT-[size] backbone.

| Model | Encoder Blocks | Decoder Blocks | Encoder Dim | Decoder Dim | Encoder Heads | Decoder Heads | Total Params | gFLOPs |
|---|---|---|---|---|---|---|---|---|
| LSM-Tiny | 4 | 2 | 192 | 128 | 3 | 4 | 2M | 0.37 |
| LSM-Small | 8 | 2 | 256 | 192 | 4 | 4 | 7M | 1.28 |
| LSM-Base | 12 | 8 | 768 | 512 | 12 | 16 | 110M | 15.94 |
| LSM-Large | 24 | 8 | 1,024 | 512 | 16 | 16 | 328M | 56.10 |

## B ADDITIONAL RELATED WORK

**Learning from Multimodal Sensor Data.** A significant body of work has explored representation learning for multimodal physiological time-series data from wearable devices. Spathis et al. (2021) employed pretext tasks to pretrain models on multimodal inputs such as heart rate and raw IMU signals, demonstrating their effectiveness across various downstream tasks. Saeed et al. (2021) introduced a self-supervised framework specifically designed for wearable sensors, emphasizing robustness through representation learning from diverse signal types. Deldari et al. (2024) proposed CrossL, a cross-modal self-supervised learning approach that utilizes latent masking to effectively model interactions between modalities. Haresamudram et al. (2021) applied contrastive predictive coding to human activity recognition, showcasing its capability to capture temporal dependencies in wearable sensor data. Further advancements include multitask learning for multi-dimensional clinical time-series (Raghu et al., 2023) and physiological measurements (Narayanswamy et al., 2024). In contrast to prior work, our work emphasizes *scalable* modeling across a broader range of wearable sensor modalities and a significantly larger data sample size. We systematically investigate scaling behavior across compute, data size, and model capacity, and examine the generalizability of the learned representations on large-scale real-world datasets.

**Self-Supervised Learning for Health.** A number of recent works have explored the utility of self-supervised pretraining as applied to health. HeAR (Baur et al., 2024) explores MAE pretraining on audio samples to embed information regarding a variety of respiratory sounds and conditions. Merrill et al.( 2023) formulate a number of pretext takes useful in embedding sensor data to enable flu and COVID-19 prediction. SimPer (Yang et al., 2023) uses a relative spectral contrastive pretraining that enables robust modeling of periodic time-series, including PPG. Most recently, PaPaGei (Pillai et al., 2025) used a PPG-morphology embedding similarity to train a PPG foundation model that generalizes to a number of clinical and consumer health tasks

## C ADDITIONAL RESULTS AND ANALYSIS

### C.1 RESULTS OF SCALING EXPERIMENTS FOR GENERATIVE TASKS

**Generative Performance wrt. Data Scaling.** Table 9 presents the full results for the generative tasks, evaluated across four model sizes and all data scales, including an experiment on the largest 40 million hour pretraining set. The `LSM` Base model, trained on 6.6 million hours of data, achieved the best overall performance.

**Scaling Pretraining Data to 40 Million Hours.** As mentioned in Sections 3 and 5, our dataset, used for the presented scaling and downstream task results, comprises 6.6M hours of data balanced across 160K people. To test the extremes of data scaling we also build a dataset comprising 40M data hours by combining the 6.6M hours with an additional 33M hours of data from a 78,569 subject subset of the total 160K subjects. However, as shown in Table 9, we observed that scaling benefits taper off when training the LSM-Base model with this extended dataset. We believe this is due to two key factors: the structure of our dataset and the inherent limitations discussed in Section 6. It is also possible that the careful balance of the 6.6M hour dataset is disturbed by the uneven distribution across subjects of the additional 33M hours.

## C.2 Results of Scaling Experiments for Discriminative Tasks

**Discriminative Performance wrt. Data Scaling.** In Table 10, we demonstrate that scaling up the dataset significantly benefits downstream discriminative tasks, particularly in the fine-tuning stage. Furthermore, our pretrained LSM model exhibits superior performance in label-efficient transfer learning, as shown in Table 11. Activity recognition few-shot results, as compared to a supervised baseline, are also visualized in Fig. 6. From the visualization it is clear that pretraining helps LSM learn a strong representation of sensor data that enables more sample efficient performance on discriminative tasks.

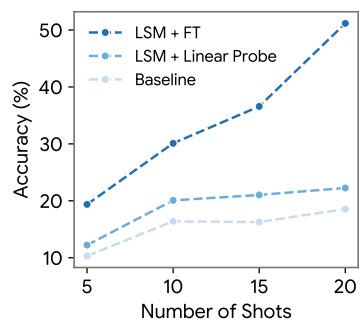

Figure 6: **Few-Shot Experiments.** Activity recognition results.

**Convolutional Probe.** Following prior work (He et al., 2022) we explore an intermediary evaluation to linear probing and full-model fine-tuning. Specifically, we explore the learnable pooling of embeddings. This probe takes patch-embeddings, produced by the encoder, and reshapes them to [num. patches $H$, num. patches $W$, embedding dimension], similar to the shape of the original patched sensor matrix. This embedding is fed through two shallow convolutional layers and a linear head. We find that with less that 0.2% of the trainable-parameters needed for full-model fine-tuning, we are able to achieve similar performance on exercise detection and activity recognition tasks. These results can be seen in Tables 10 and 11.

## C.3 Classification Confusion Matrices

Fig. 7 presents the complete confusion matrix for our activity recognition task from the full-model fine-tuned LSM-B model. Note that many classes get mistaken for *Walk*. This is likely as there are significant periods of walking in the 5-hour inputs, even if the activity is labeled otherwise.

## C.4 Feature Embeddings

We present t-distributed Stochastic Neighbor Embedding (t-SNE) plots. Fig. 8 illustrates that scaling pretraining data results in noticeable, albeit subtle, improvements of clustering across activities in the learned representation. We also find that fine-tuning the model is critical to effectively discriminate between activities. In Fig. 9 we see that the learned representation does embed some subject dependencies. This can be attributed to variance in the physiology and activity definitions of individuals (e.g., a hard run may look very different for two different people). We present quantitative results of a subset of these embeddings in Table 12.

## C.5 Signal Correlations

The 26 signals used as input to our model are derived from five sensors (PPG, accelerometer, EDA, temperature, and altimeter). Consequently, some features exhibit stronger correlations with specific others. A signal diagonal correlation matrix was calculated to show the pairwise correlations between signals. Fig. 10 shows the correlation matrix for signals clustered by sensor and for signals ordered to minimize the absolute correlation coefficient between adjacent features.

Table 9: **Effect of Scaling on Generative Tasks (Full Results).** Data size is in hours.

| Data Size | Model Size | Task Error (MSE) | | |
| --- | --- | --- | --- | --- |
| | | Random Imputation 80% | Extrapolation 60 min | Interpolation 60 min |
| 0.005 M | Tiny | 0.50 | 0.71 | 0.53 |
| | Small | 0.57 | 0.77 | 0.62 |
| | Base | 0.67 | 0.80 | 0.68 |
| | Large | 0.64 | 0.82 | 0.75 |
| 0.05 M | Tiny | 0.25 | 0.70 | 0.47 |
| | Small | 0.29 | 0.58 | 0.36 |
| | Base | 0.38 | 0.65 | 0.42 |
| | Large | 0.38 | 0.65 | 0.43 |
| 0.5 M | Tiny | 0.22 | 0.62 | 0.42 |
| | Small | 0.21 | 0.53 | 0.37 |
| | Base | 0.22 | 0.48 | 0.28 |
| | Large | 0.22 | 0.50 | 0.34 |
| 3.8 M | Tiny | 0.22 | 0.62 | 0.42 |
| | Small | 0.21 | 0.49 | 0.36 |
| | Base | **0.19** | **0.44** | **0.26** |
| | Large | 0.21 | 0.64 | 0.46 |
| 6.6 M | Tiny | 0.22 | 0.63 | 0.42 |
| | Small | 0.21 | 0.49 | 0.35 |
| | Base | **0.19** | **0.44** | **0.26** |
| | Large | 0.20 | 0.54 | 0.40 |
| 40 M | Base | **0.19** | 0.45 | 0.27 |

Table 10: **Effect of Data Scaling on Discriminative Tasks.** Data size is in hours.

| Data Size | Method | Exercise Detection | | Activity Recognition | |
| --- | --- | --- | --- | --- | --- |
| | | Accuracy | mAP | Accuracy | mAP |
| 0.005 M | | 60.6 | 49.8 | 35.1 | 17.5 |
| 0.05 M | | 67.3 | 61.0 | 39.6 | 23.4 |
| 0.5 M | Linear Probe | 84.5 | 78.8 | 47.1 | 24.7 |
| 3.8 M | | 88.0 | 85.0 | 47.6 | 25.3 |
| 6.6 M | | 84.7 | 89.0 | 49.4 | 24.6 |
| 0.005 M | | 71.3 | 71.8 | 50.9 | 25.1 |
| 0.05 M | | 78.0 | 82.3 | 62.2 | 43.7 |
| 0.5 M | Convolutional Probe | 88.2 | 96.4 | 68.1 | 45.5 |
| 3.8 M | | 88.2 | 96.4 | 70.5 | 47.1 |
| 6.6 M | | 87.5 | 95.8 | 67.6 | 48.5 |
| 0.005 M | | 68.3 | 58.9 | 51.5 | 30.0 |
| 0.05 M | | 73.8 | 77.0 | 64.0 | 48.0 |
| 0.5 M | Fine-Tune | 84.9 | 93.7 | **68.8** | 50.0 |
| 3.8 M | | 87.5 | 96.4 | 64.2 | 48.7 |
| 6.6 M | | **90.3** | **97.0** | 68.5 | **51.4** |

## C.6 EXAMPLES OF RECONSTRUCTIONS

A qualitative example of ground-truth signals and corresponding reconstructions are shown in Fig. 11. The gray regions are sections that were masked in the input. Additional sensor matrix level reconstructions, across generative down-stream tasks (e.g., imputation, extrapolation, interpo-

Table 11: **Few-Shot Performance on Discriminative Tasks.**

| Samples per Class | Method | Exercise Detection | | Activity Recognition | |
|---|---|---|---|---|---|
| | | **Accuracy** | **mAP** | **Accuracy** | **mAP** |
| 5 | | 51.3 | 48.0 | 12.2 | 17.5 |
| 10 | Linear Probe | 58.3 | 57.1 | 20.1 | 18.4 |
| 15 | | 65.4 | 68.8 | 21.0 | 18.7 |
| 20 | | 65.1 | **69.8** | 22.3 | 18.8 |
| 5 | | 40.5 | 43.8 | 20.6 | 24.7 |
| 10 | Convolutional Probe | 63.2 | 59.4 | 27.9 | 26.7 |
| 15 | | 57.3 | 60.8 | 27.9 | 26.7 |
| 20 | | **67.0** | 56.9 | 36.9 | 25.3 |
| 5 | | 54.7 | 56.8 | 19.4 | 21.5 |
| 10 | Fine-Tune | 65.8 | 65.1 | 30.1 | 22.7 |
| 15 | | 71.1 | 73.1 | 36.6 | 24.8 |
| 20 | | 65.6 | 67.1 | **51.2** | **33.2** |
| 5 | | 43.1 | 52.9 | 10.3 | 14.5 |
| 10 | Supervised | 49.3 | 46.0 | 16.4 | 14.6 |
| 15 | | 49.6 | 50.6 | 16.3 | 14.4 |
| 20 | | 48.2 | 45.8 | 18.5 | 23.0 |

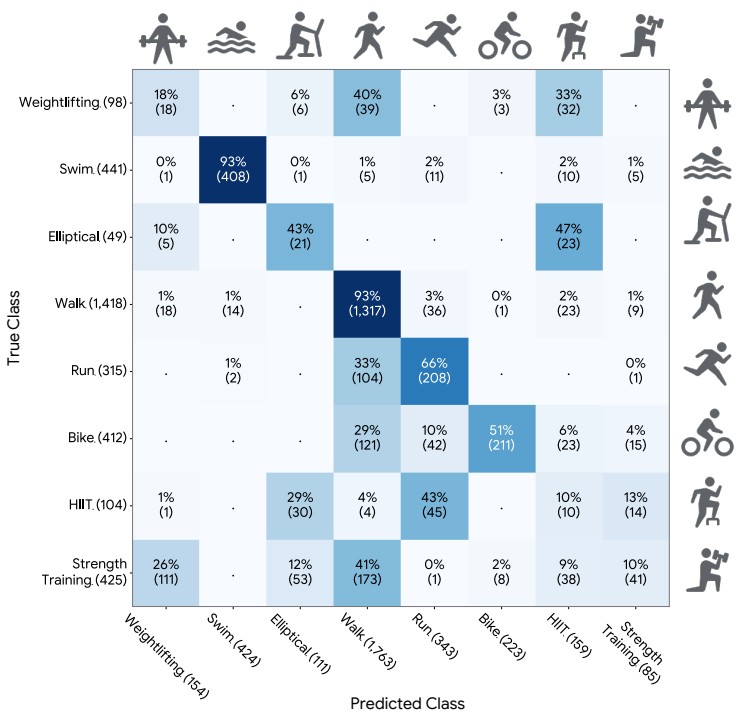

Figure 7: **Activity Recognition Confusion Matrix** for the full-model fined-tuned LSM-Base.

lation) can be seen in Fig. 12. Examples of the effect of scaling on reconstruction can be seen in Fig. 13. Note that these effects are often visually subtle (e.g., signal smoothness, patch aberrations, etc.)

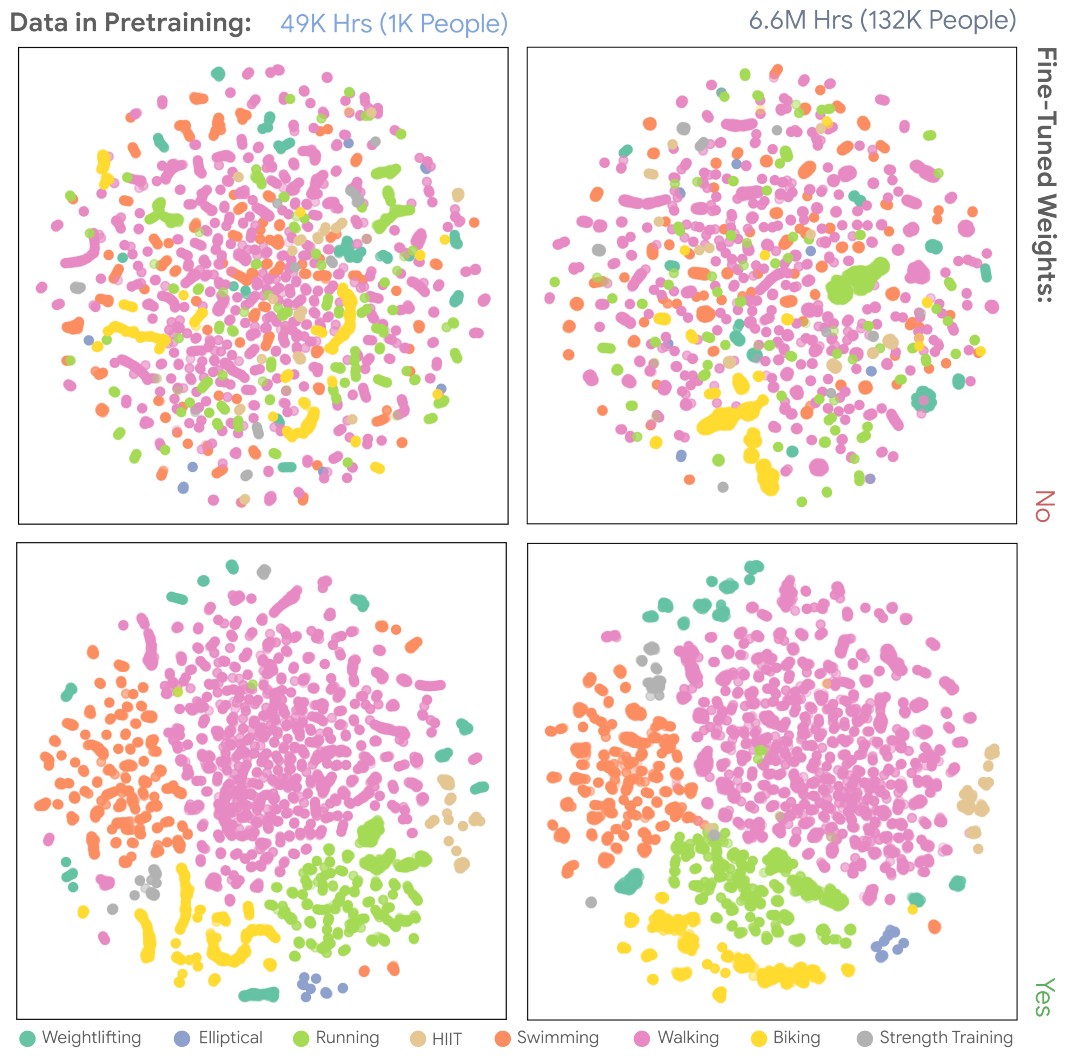

Figure 8: **t-SNE Embeddings for Pretraining and Fine-Tuned Models Labeled by Activity.** t-distributed Stochastic Neighbor Embedding (t-SNE) plots showing that there are differences (albeit subtle) between pretrained embeddings using data from almost 50k and 6.6M hours.

Table 12: **Measures of Embedding Clustering**. These quantitative evaluations of clustering support the claim that there are subtle differences in embeddings pretrained on different scales of data, as visualized in Fig. 8.

| Fine-tune | Pretrain Size (Hrs) | Pretrain Size (People) | Calinski-Harabasz ↑ | ARI ↑ | NMI ↑ |
|-----------|--------------------|-----------------------|---------------------|-------|-------|
| No | 49K | 1K | 11508 | 0.026 | 0.083 |
| No | 6.6M | 132K | 11855 | 0.034 | 0.113 |
| Yes | 49K | 1K | 11972 | **0.256** | 0.465 |
| Yes | 6.6M | 132K | **11993** | 0.253 | **0.501** |

Calinski-Harabasz Score (Caliński & Harabasz, 1974), Adjusted Rand Index (ARI) (Hubert & Arabie, 1985), Normalized Mutual Information Score (NMI) (Vinh et al., 2009).

## C.7 ADDITIONAL DISCRIMINATIVE TASKS

Here we discuss the additional discriminative tasks of sex and age classification, and subject dependent mood recognition. The datasets for these tasks are presented in Table 13. The performance of our methods and baselines are presented in Table 14.

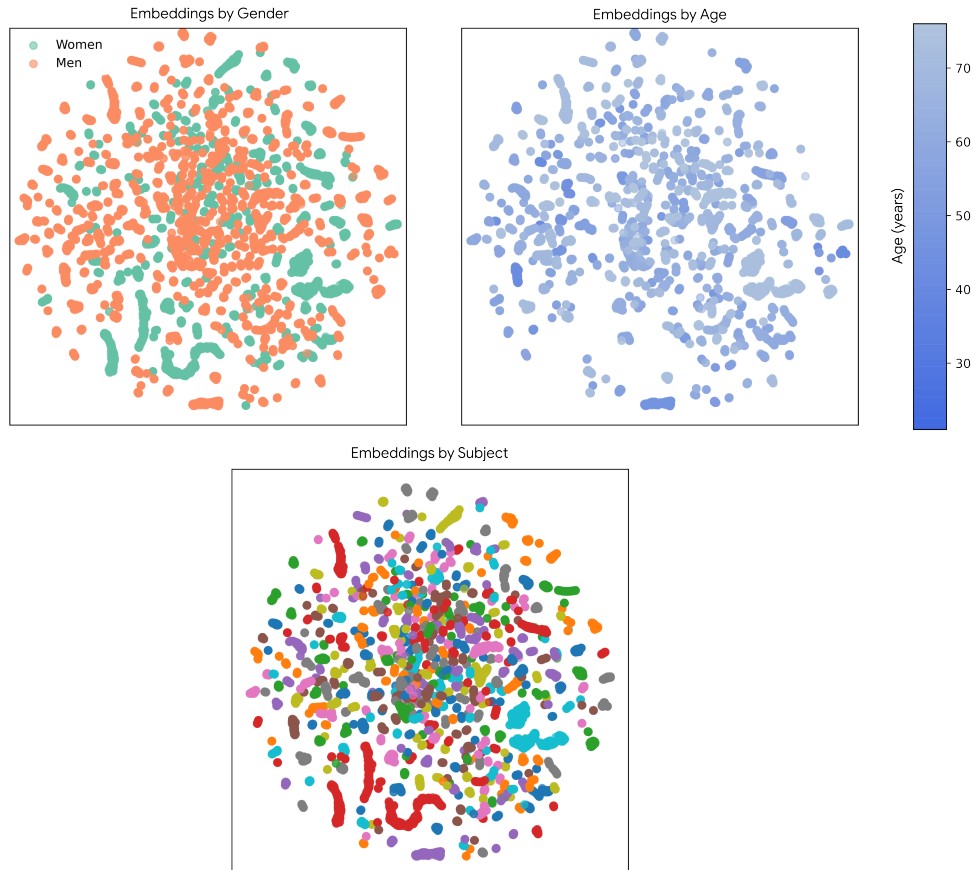

Figure 9: **t-SNE Embeddings Labeled by Gender, Age and Subject.** t-distributed Stochastic Neighbor Embedding (t-SNE) plots showing that the learned embeddings do capture subject specific information (and therefore also exhibit some subtle gender and age clusters). Age was not available for all subjects.

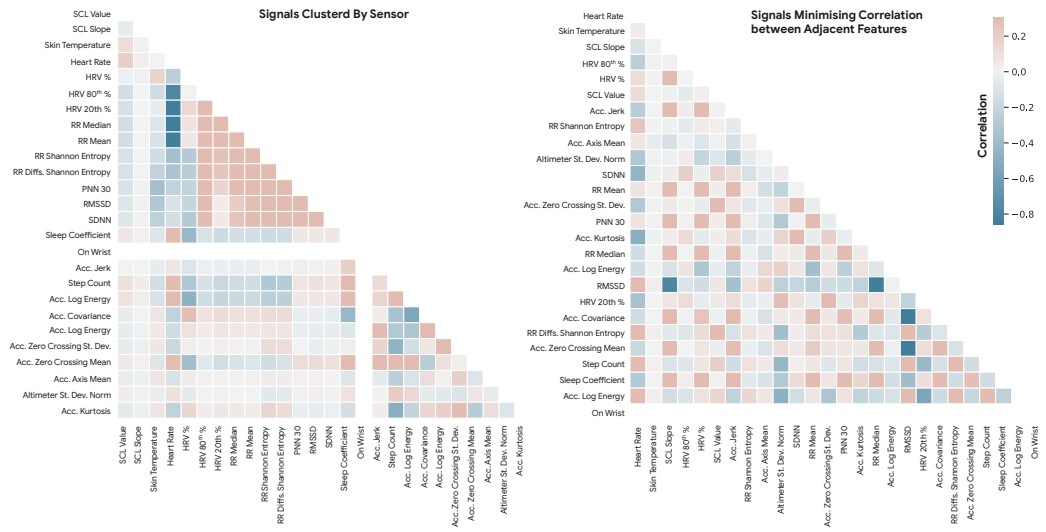

Figure 10: **Sensor Signal Diagonal Correlation Matrix.** The pair-wise correlation between the 26 sensor features as based on our training set.

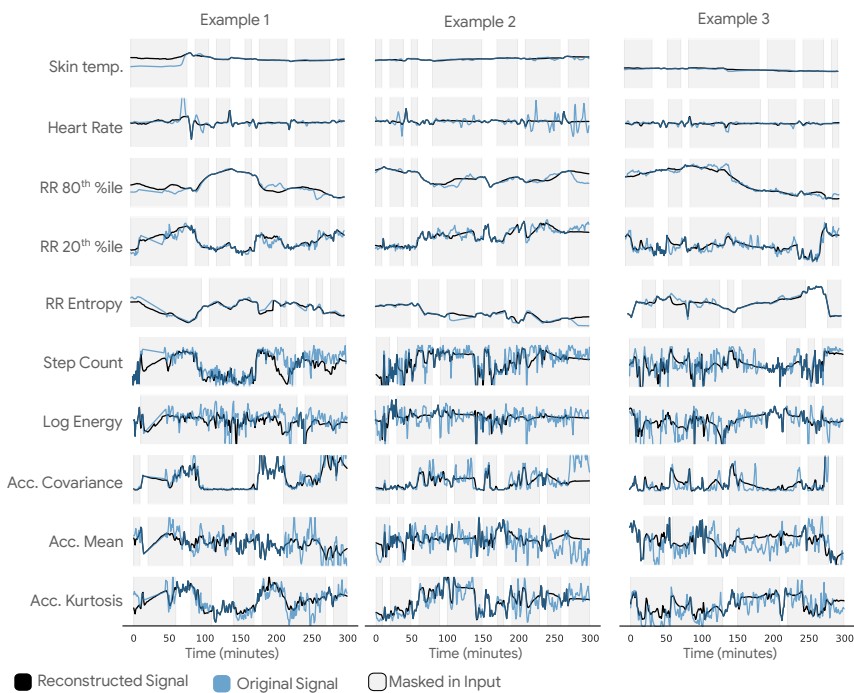

Figure 11: **Examples of Signal Reconstructions.** Comparison between the ground-truth (blue) and reconstruction (black) for a 5-hour sample. Gray regions were masked in the input. 80% Random Masking (Patch Size 10 mins x 5 sensors). Note: model outputs are only shown for the masked regions in the reconstructions.

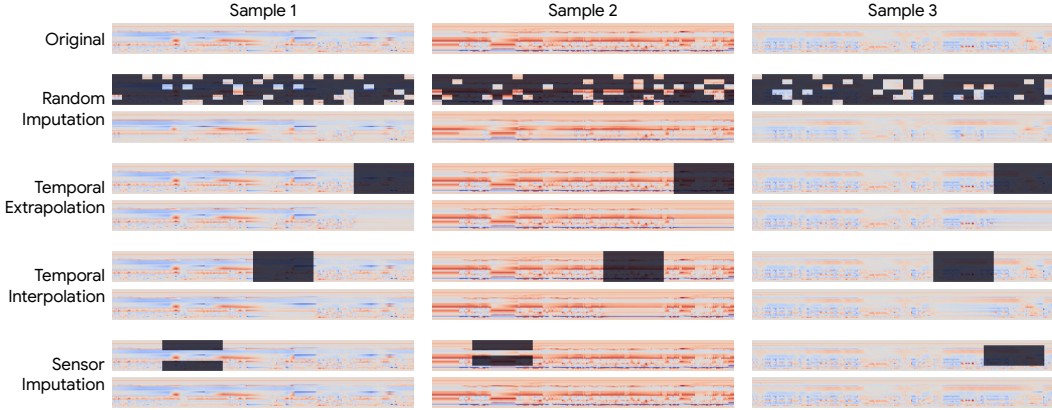

Figure 12: **Examples of Signal Reconstructions Across Generative Downstream Tasks.** The top row of each sample shows the original sensor signal matrix (visualized as an image). Subsequent row-pairs plot the masked input followed by the model reconstruction below. All reconstruction come from LSM-Base employing a 10x5 (minutes x features) patch size and pretrained with 80% random masking. Note: model outputs are only shown for the masked patches in the reconstructions.

**Biological Sex Classification.** Biological sex classification attempts to categorize an individual as either *Female*, or *Male* from a sample of their wearable data. Though we do not explicitly intend our learned representation to encode this information, this discriminative task may allude to a method's ability to understand differences in human biology and physiology. This experiment mirrors those conducted by similar work (Abbaspourazad et al., 2023).

**Binned Age Classification.** Binned age classification defines 4 discrete age buckets. These binned ranges are [18 - 34], [35 - 49], [50 - 64], 65+. Although our pretraining methods do not explicitly

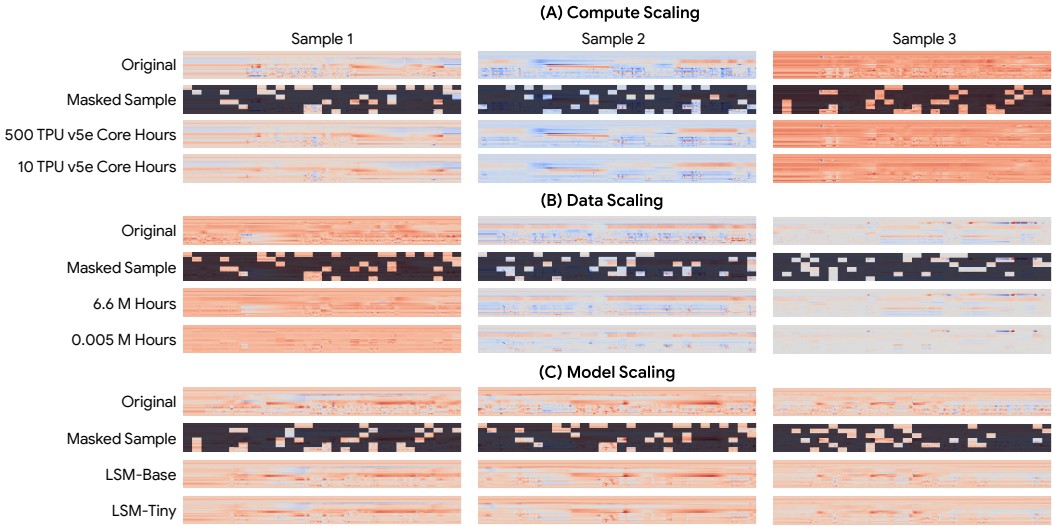

Figure 13: **Examples of Signal Reconstructions with Respect to Scaling.** These plots illustrate the (often visually subtle) effect of **(A)** compute, **(B)** data, and **(C)** model scaling for sensor models. Note: model outputs are only shown for the masked patches in the reconstructions.

intend to learn age, there are known correlations between a person's age and their physiological state (Cheitlin, 2003). To this end, the task of binned age classification provides a proxy signal to understand the extent to which a method is able to model differences in physiology. This experiment mirrors those conducted by related work (Abbaspourazad et al., 2023).

**Subject Dependent Mood Recognition.** Mood recognition is the processes of categorizing a person's emotion events given patterns detected in their sensor data. Such mood events may cause changes in the activity of the autonomic nervous system which then result in measurable physiological changes (Ekman et al., 1983). Our dataset includes five user-labeled mood states: *Content, Frustrated, Excited, Calm, Stressed*. We find, however, that these mood events vary significantly from person to person. This is likely due to the innate ambiguity associated with people's emotions. For example, one person's *Frustration* may easily be construed as *Stress* by another. Motivated by this, we focus on the task of subject dependent mood recognition, where for a given person their mood event samples are split 80-20 between train-test sets.

Table 13: **Details of Additional Discriminative Datasets.** Summary of class distributions for our additional discriminative task datasets.

| Task | Class | # Training | # Testing |
|------|-------|-----------|-----------|
| **Sex** | Female | 9,181 | 2,160 |
| | Male | 5,022 | 1,090 |
| | **Total** | 14,203 | 3,250 |
| **Age** | 18-35 | 1,179 | 499 |
| | 35-49 | 5,697 | 793 |
| | 50-64 | 5,246 | 1,297 |
| | $\geq 65$ | 2,250 | 673 |
| | **Total** | 14,372 | 3,262 |
| **Mood** | Content | 786 | 238 |
| | Frustrated | 460 | 199 |
| | Excited | 408 | 118 |
| | Calm | 767 | 265 |
| | Stressed | 1,132 | 334 |
| | **Total** | 3,553 | 1,154 |

Table 14: **Comparisons of LSM and Baseline Methods on Additional Discriminative Tasks.** Tasks include biological sex classification, binned age classification, and subject dependent mood recognition.

| Pretrain | Probe/FT | Sex Classification | | Age Classification | | Mood Recognition | |
|---|---|---|---|---|---|---|---|
| | | Acc. | mAP | Acc. | mAP | Acc. | mAP |
| - | RANDOM FOREST | 66.2 | 56.6 | **42.3** | 35.3 | 41.8 | **44.4** |
| - | LOGISTIC REGRESSION | 59.9 | 54.0 | 28.9 | 28.5 | 43.5 | 42.8 |
| - | SUPERVISED | 66.0 | 60.1 | 35.3 | 33.4 | 42.4 | 32.7 |
| MSN | Linear Probe | 68.0 | 61.9 | 36.4 | 29.2 | 36.7 | 34.8 |
| DINO | Linear Probe | 66.4 | 64.6 | 31.0 | 29.9 | 40.6 | 36.5 |
| SIMCLR | Linear Probe | 67.6 | 60.3 | 37.6 | 26.3 | 36.2 | 30.1 |
| LSM (MAE) | Linear Probe | **75.4** | 75.3 | 40.9 | 38.2 | 42.0 | 34.7 |
| MSN | Fine-tune | 68.3 | 63.8 | 43.8 | 36.6 | 41.2 | 38.8 |
| DINO | Fine-tune | 67.9 | 60.9 | 36.9 | 34.4 | 43.5 | 37.9 |
| SIMCLR | Fine-tune | 65.2 | 57.6 | 37.3 | 33.5 | 43.8 | 37.4 |
| LSM (MAE) | Fine-tune | **75.4** | 79.8 | 42.2 | **38.7** | **49.0** | 43.0 |
| GAIN OVER SUPERVISED | | **+14.1%** | **+32.8%** | **+19.5%** | **+15.9%** | **+15.6%** | **+31.5%** |

All neural methods, including the supervised method, utilize a ViT-Base (110M) backbone. Relevant methods are pretrained with 6.6 million hours of data. In the sensor imputation task, we randomly mask 67% of the sensor modalities. Random Forest (Breiman, 2001), ViT (Dosovitskiy et al., 2021), MSN (Assran et al., 2022), DINO (Caron et al., 2021), SimCLR (Chen et al., 2020), MAE (He et al., 2022).

# D  ADDITIONAL DISCUSSIONS, LIMITATIONS, AND FUTURE WORK

## D.1  DISCUSSIONS

**Can Wearable Foundation Models Scale to Edge Devices?** Our model saturation at 110 million parameters highlights a practical advantage: models of this size can potentially run in real-time on modern mobile devices, leveraging advancements in on-device large language models that often exceed 1 billion parameters in on-device deployments (Xu et al., 2024). We further note that the strong performance of statistical machine learning baselines, on classification tasks, indicate that there may be utility in distilling LSM to be smaller and more edge-performant. Additionally, unlike prior research focused on raw sensor data (e.g., (Yuan et al., 2024)), our results establish that scaling is effective with per-minute aggregated data. This approach not only reduces privacy risks but also introduces a standardized format that is easier to unify across platforms and devices. Looking forward, these findings open the door to collaborative efforts such as federated learning (Lim et al., 2020) across different wearable manufacturers, enabling scalable, privacy-preserving training, while maintaining compatibility across diverse hardware ecosystems.

**Contextualizing Wearable Foundation Models.** As highlighted by Bommasani et al. (2021), foundation models are often characterized by their training on vast and diverse data (data scaling), and their use of large-scale architectures (model scaling), which jointly enable the robust understanding of general domain features and adaptability to a breadth of downstream tasks. Bommasani et al. (2021) further emphasize that transfer learning underpins the versatility of these models, while scaling amplifies their power. Additionally, while some foundation models exhibit multimodal capabilities, this is not a universal requirement; for example, early GPT models (Brown et al., 2020) focus on text, and models like DALL-E (Ramesh et al., 2021) specialize in images. Handling multiple devices, similarly, should be viewed as an application-specific extension rather than a core defining characteristic. This work, on scaling laws in the domain of multimodal wearable data, aligns with these principles, along with those of relevant prior work (Abbaspourazad et al., 2023), and sets a foundation for future studies regarding the application of such models across more diverse devices, sensor arrays, and downstream tasks.

**Understanding Demographic Proxy Tasks.** The ability of a foundation model to predict demographic information does not necessarily imply that the model is explicitly designed to capture this information (Vaidya et al., 2024; Yang et al., 2024). As shown in prior work, even randomly initialized neural networks can predict demographic attributes such as age or gender in medical imaging contexts (e.g., chest X-rays) without targeted design (Glocker et al., 2023). Although we do not explicitly train our methods to encode demographic information, proxy tasks, such as predicting age or biological sex, shed light on a models ability to understand physiological characteristics relevant to general health. Similar analyses have been conducted in prior work (Abbaspourazad et al., 2023).

The ability of these methods to encode such information may raise privacy concerns. To this end, we again emphasize that our models were not explicitly trained to encode demographic information, and the observed results of in Table C.7 are modest even with task specific fine-tuning. Furthermore, we postulate that our minutely aggregate features help preserve privacy, as compared with prior works (Abbaspourazad et al., 2023) which use raw sensor feeds and are able to more robustly predict demographic information.

We acknowledge that health and demographic information are sensitive, and thus should be studied with consideration. It is well studied that machine learning models, in various domains including health, exhibit biases (Mehrabi et al., 2021; Vaidya et al., 2024). Understanding these biases is paramount to addressing them and making these methods more equitable and transparent. Proxy tasks, that evaluate a model's performance on splits of demographic information, shed light onto the performance of a model for different groupings of people. To this end we refer the reader to prior works that dive deep into the associated risk and importance of conducting such investigations (Gichoya et al., 2022; Glocker et al., 2023).

### D.2 LIMITATIONS AND FUTURE WORK

**The Effect of Missing Data on Discriminative Tasks.** As previously mentioned, missingness is inherent in wearable sensor data. This may occur due to sporadic, unplanned events such as battery depletion, environmental factors (e.g., extreme temperatures), or due to planned events such as charging one's device at night. To this end our current evaluation utilizes missingness during MAE pretraining. However, we do not explore the ability of missing data to affect downstream classification tasks. Future work may explore the tolerance of LSM like models to missingness and the ability of these models to provide accurate classifications based on incomplete data.

**Generalizing to New Devices and Sensor Configurations.** We want to acknowledge that our methods are limited in that they focus on a narrow subset of wearable devices and corresponding sensors. As wearables continue to increase in their ubiquity, more work is needed to explore how models, like LSM, can be adapted for consumer hardware with different form factors, firmware, and sensor configurations. Specifically, our work begs the questions: How can wearable foundation models generalize to new sensor modalities? How can these models generalize to distribution shifts caused by different sensor hardware and firmware? How can these models adapt to provide robust performance in the absence of some or many sensor feeds?

## E ADDITIONAL IMPLEMENTATION DETAILS

### E.1 HYPERPARAMETERS

This section provides details about the pretraining and fine-tuning of LSM and other baseline methods. The pretraining hyperparameters, detailed in Table 15, were chosen with hyperparameter sweeps. In Table 16, we include hyperparameters for linear probe and fine-tuning. The hyperparameters for supervised baseline training are detailed in Table 17. Note that hyperparameters used for the few-shot experiments found in Table 11 are similar to those found in Tables 16 and 17 with slight changes in learning rate.

### E.2 TRAINING AUGMENTATIONS

Traditional image augmentations are not always valid when applied to sensor matrices. For example, random crop and resize, often applied in contrastive pretraining is invalid for sensor matrices, as a random crop may remove a subset of sensor signals. Thus, we define a subset of augmentations valid for sensor matrices. These are random *Flip*: a flip along the temporal axis; *Stretch*: a stretch along the temporal axis and an subsequent crop back to original time length; and *Noise*: the addition of Gaussian noise.

### E.3 DISCRIMINATIVE TASK TUNING AND EVALUATION

In an effort to mitigate confusion regarding the implementation of our downstream discriminative tasks we cover possible questions and implementation details. All discriminative tasks operate on 5-hour wearable sensor feeds, the same form of input used during pretraining. These windowed

samples are associated with metadata including a person's biological sex, age, and any activity events occurring during the 5 hour span. Activity labels are self-reported with the corresponding events varying in duration (e.g., a bike ride may last several hours, whereas a HIIT workout may only last 30 minutes). Generally, these events are self-reported post hoc. As activity events are self-reported it is possible that other unreported events may also exist in the 5 hour span. As we discuss in Appendix C.3, this may result in degraded performance.

Table 15: **Hyperparameters for Pretraining** with MAE (He et al., 2022), MSN (Assran et al., 2022), DINO (Caron et al., 2021) and SimCLR (Chen et al., 2020). A solitary row value indicates that the value was used for all methods.

| Configuration | LSM MAE | MSN | DINO | SimCLR |
|---|---|---|---|---|
| Training Steps | \multicolumn{4}{c}{50,000} | | | |
| Warmup Steps | 2,500 | | | |
| Optimizer | AdamW (Loshchilov & Hutter, 2017) | | | |
| Opt. momentum $[\beta_1, \beta_2]$ | [0.9, 0.95] | [0.9, 0.99] | [0.9, 0.99] | [0.9, 0.99] |
| Base learning rate | 0.005 | 0.001 | 0.004 | 0.001 |
| Batch size | 4,096 | | | |
| Weight decay | 0.0001 | | | |
| Gradient clipping | 1.0 | 3.0 | 3.0 | 3.0 |
| Dropout | 0.0 | | | |
| Learning rate schedule | Linear Warmup & Cosine Decay | | | |
| Loss Function | Mean Squared Error | | | |
| Data resolution | 26 (signal)×300(minute) | | | |
| Augmentation | Flip, Stretch, Noise | | | |

Table 16: **Hyperparameters for LSM Linear Probing and Fine-Tuning on Discriminative Tasks** detailed in Section 4.2. A solitary row value indicates that it was used for all methods. LP=Linear Probe. FT=Fine-Tune (full model).

| Task | Exercise Detection | | Activity Recognition | | Sex Classification | | Age Classification | | Mood Recognition | |
|---|---|---|---|---|---|---|---|---|---|---|
| Configuration | LP | FT | LP | FT | LP | FT | LP | FT | LP | FT |
| Training Steps | 400 | 400 | 300 | 300 | 300 | 300 | 300 | 300 | 300 | 300 |
| Warmup Step Percent | 20 | 20 | 15 | 15 | 15 | 15 | 15 | 15 | 15 | 15 |
| Optimizer | AdamW (Loshchilov & Hutter, 2017) | | | | | | | | | |
| Opt. momentum $[\beta_1, \beta_2]$ | [0.9, 0.95] | | | | | | | | | |
| Base learning rate | 0.5 | 0.00005 | 0.5 | 0.00005 | 0.05 | 0.0005 | 0.05 | 0.00005 | 0.005 | 0.005 |
| Batch size | 128 | 128 | 128 | 128 | 128 | 128 | 256 | 256 | 256 | 256 |
| Weight decay | 0.0001 | | | | | | | | | |
| Gradient clipping | 1.0 | | | | | | | | | |
| Dropout | 0.3 | 0.3 | 0.3 | 0.3 | 0.1 | 0.3 | 0.1 | 0.1 | 0.7 | 0.1 |
| Learning rate schedule | Linear Warmup & Cosine Decay | | | | | | | | | |
| Loss Function | Balanced Softmax Loss (Ren et al., 2020) | | | | | | | | | |
| Data resolution | 26 (signal)×300(minute) | | | | | | | | | |
| Augmentation | Noise | | | | | | | | | |

Table 17: **Hyperparameters for Supervised Training on Discriminative Tasks.** A solitary row value indicates that it was used for all methods.

| Configuration | Exercise Detection | Activity Recognition | Sex Classification | Age Classification | Mood Recognition |
|---|---|---|---|---|---|
| Training Steps | 400 | 300 | 300 | 300 | 300 |
| Warmup Steps | 20 | 15 | 15 | 15 | 15 |
| Optimizer | AdamW (Loshchilov & Hutter, 2017) | | | | |
| Opt. momentum $[\beta_1, \beta_2]$ | [0.9, 0.95] | | | | |
| Base learning rate | 0.0001 | 0.0005 | 0.005 | 0.0001 | 0.00005 |
| Batch size | 128 | 128 | 128 | 256 | 256 |
| Weight decay | 0.0001 | | | | |
| Gradient clipping | 1.0 | | | | |
| Dropout | 0.0 | | | | |
| Learning rate schedule | Linear Warmup & Cosine Decay | | | | |
| Loss Function | Balanced Softmax Loss (Ren et al., 2020) | | | | |
| Data resolution | 26 (signal)×300(minute) | | | | |
| Augmentation | Noise | | | | |

## F  DESCRIPTION OF PRETRAINING AND BASELINE METHODS

### F.1  PRETRAINING METHODS

There are two main approaches to pretraining. (a) Contrastive methods learn similarities between different views of the same example (*positives*) and differences when compared to other examples (*negatives*). (b) Reconstruction tasks focus on re-generating the input after it has been distorted by some function. Although both are popular, there are compelling arguments as to why reconstructive tasks may be useful for sensor modeling. First, in the sensor domain it can be non-trivial to create augmentations that do not alter the meaning (label) of the sample. For example, does stretching data for someone *running* mean that it more closely resembles the data when they *walk*? Second, generative capabilities are attractive as imputing missing data and forecasting signals into the future are useful in and of themselves. Motivated by its generative capabilities and its proven scaling in other domains (e.g., vision, audio), we select a masked autoencoder (He et al., 2022) as the candidate method for LSM.

**Masked Auto Encoder (MAE) (He et al., 2022).** MAE is a self-supervised learning method where the input data is randomly masked, and the model is trained to reconstruct the missing parts. It operates on the principle that forcing the model to predict missing information helps it learn meaningful representations. MAE has shown strong performance in various vision and signal tasks, particularly in cases where large-scale unlabeled data is available.

**SimCLR (Chen et al., 2020).** SimCLR is a contrastive learning framework that learns representations by maximizing agreement between different augmented views of the same data sample. The method uses a contrastive loss, which encourages the model to pull together similar views of the same sample while pushing apart views of different samples. SimCLR has been widely used with both visual and sensor data for representation learning without requiring labeled data.

**Masked Siamese Network (MSN) (Assran et al., 2022).** MSN combines the benefits of invariance-based pretraining with mask denoising. MSN operates by matching the representation of an image view with randomly masked patches to the representation of the corresponding unmasked image. This pretraining strategy leverages Vision Transformers by processing only the unmasked patches, significantly enhancing scalability. The framework enables the generation of semantically rich representations, which perform competitively in low-shot image classification tasks.

**DINO (Caron et al., 2021).** DINO is a self-distillation method that trains the model using knowledge distillation, without the need for labeled data. It leverages a teacher-student network architecture, where the teacher generates target representations for the student to learn from. DINO has demonstrated success in generating robust representations that can be transferred to various downstream tasks.

### F.2  GENERATIVE BASELINES

We define a number of baselines for our generative tasks. Similar methods are common-place in the image domain (often used for up-sampling) (Han, 2013), and the Internet of Things (IoT) sensor domain (often for imputing corrupted and/or missing data) (Adhikari et al., 2022).

**Mean Fill.** Mean Fill is a simple baseline for generative tasks, where the missing values for a sensor stream are replaced by the mean value of the sensor data present in a given sample. Though naive, this method provides a reasonable estimate in certain contexts where missing values are randomly distributed.

**Nearest Neighbor Fill.** Nearest Neighbor Fill imputes missing data by using the value of the nearest observed neighbor for a given feature along the temporal axis. In the absence of past and future neighbors this method mirrors back/forward fill. This method works well when there is a high degree of local similarity in the data.

**Linear Interpolation.** Linear Interpolation fills missing values by interpolating linearly between known values along the temporal dimension. In the absence of past and future neighbors this method mirrors back/forward fill. This baseline is often used for time-series and spatial data, where the assumption is that changes between data points occur in a smooth, continuous manner.

**Multivariate Imputation by Chained Equations (MICE) (Van Buuren & Groothuis-Oudshoorn, 2011).** MICE is an imputation technique which builds a regression model, per feature, conditional upon all observed data across all variables. The method iterates through feature variables and imputes missing data. This iteration process is run multiple times with the imputation being refined each iteration. It should be noted that MICE performs best when missingness is random (Azur et al., 2011) and may under-perform in situations of large or structured missing data.

For all generative baseline methods, in the rare cases where the sensor feature is completely missing, the feature values are replaced with zeros. This remains a valid strategy as all features are z-score normalized and centered around zero.

### F.3 CLASSIFICATION BASELINES

**Vision Transformer (ViT) (Dosovitskiy et al., 2021).** The Vision Transformer (ViT) is a transformer-based architecture that treats image patches or signal segments as input tokens, similar to how transformers handle sequences in natural language processing. ViT has shown competitive performance across various classification tasks, especially when trained with large amounts of data, and serves as a strong baseline in both vision and sensor classification tasks.

**Random Forest (Breiman, 2001).** Random forest is a statistical machine learning method. This ensemble method fits multiple decision trees to various splits of a dataset. When applied to a classification task, the random forest uses a majority voting scheme across the decision trees to classify the sample.

**Logistic Regression Classifier.** A logistic regression model applies the sigmoid function to a linear combination of input variables to predict some output value. In the case of a $N$ multi-class classification problem $N$ logistic regression models are created, each modeling binary classification of a single class. The resulting $N$ model probability scores are compared with the highest score dictating the predicted class.

## G ADDITIONAL DETAILS OF DATASET

### G.1 DATASET DETAILS

In Table 18 we detail the 26 derived sensor signal features leveraged by our methods.

### G.2 PRACTICAL CONSTRAINTS OF DATA COLLECTION

As mentioned earlier in Section 3.1, we use 1-minute granularity features due to practical constraints. Specifically these features are aggregated at the minute level and saved on-device at this minute granularity. The exception is heart rate which is calculated per second. This is done to reduce the battery and memory burden of writing and saving raw features. Furthermore, streaming data from the wearable device presents another practical constraints, and the use of aggregated features allows easier transfer of this data.

### G.3 THE UTILITY OF MINUTELY FEATURES

Raw sensors are generally sampled at much higher rate than our minutely features. Through this compression it is likely that we lose features vital in capturing higher frequency activities. For this reason it is important to ground the utility of our minutely features and their ability to discriminate different life events. To this end we add a qualitative example of minutely wearable data across different activities aggregated across samples. Fig. 14 indicates that there are differences, albeit subtle, between activities at the minutely resolution.

### G.4 SAMPLING OF EVENT CLASSES

Event labels, for tasks like activity and mood recognition, do not occur naturally balanced. For this reason, we sub-sample event data to generate more balanced task-specific datasets. For example almost 90% of logged activities are walking, and thus the activities are sampled to increase the relative

proportion of under-represented classes (e.g., elliptical and weightlifting). The natural distribution of these events along with our sampled distributions can be found in Table 19.

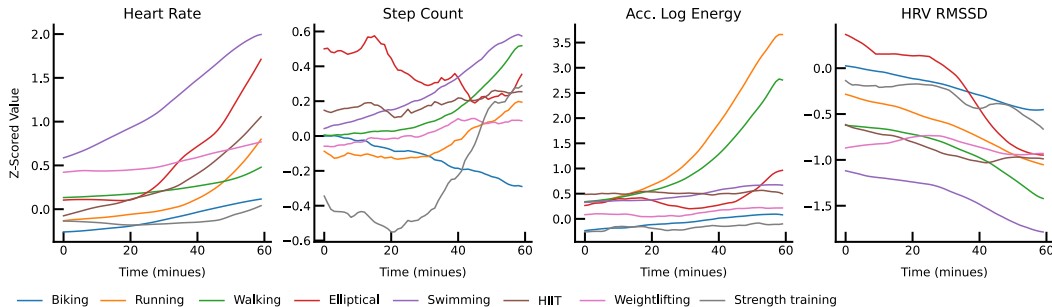

Figure 14: **Average Features by Activity Class.** Average (i) Heart Rate, (ii) Step Count, (iii) Accelerometer Log Energy and (iv) HRV RMSSD for each of the activity classes.

Table 18: **Sensor Feature Definitions.** Names, units, and definitions of the 26 PPG, accelerometer, skin conductance, skin temperature, and altimeter features we use.

| Feature | Unit | Definition |
|---|---|---|
| **PPG** **Photoplethysmography** | | |
| Heart Rate | Beats/Min | Mean of instantaneous heart rate. |
| RR Percent Valid | % | % of 5-minute window with valid RR intervals. |
| RR $80^{th}$ Percentile | Msec | $80^{th}$ percentile of 5-minute window of RR ints. |
| RR $20^{th}$ Percentile | Msec | $20^{th}$ percentile of RR ints. |
| RR Median | Msec | Median RR interval. |
| RR Mean | Msec | Mean RR interval. |
| Shannon Ent. RR | Nats | Shannon entropy of the RR intervals. |
| Shannon Ent. RR Diffs | Nats | Shannon entropy of the RR interval differences. |
| PNN30 | % | % of successive RR ints. that change by $> 30$ms. |
| RMSSD | Msec | Root mean squared st. dev. of RR ints. |
| SDNN | Msec | Standard deviation of RR intervals. |
| On Wrist | Boolean | If optical-sensor off-wrist within a 30-second window, then false. |
| **ACC** **Accelerometer** | | |
| Jerk Autocorrelation Ratio | a.u. | Ratio of lag=1 autocorrelation to energy in 1st 3-axis principal component. |
| Step Count | Steps | Number of steps. |
| Log Energy | a.u. | Log of sum of 3-axis root mean squared magnitude. |
| Covariance Condition | a.u. | Estimate of condition number for 3-axis covariance matrix. |
| Log Energy Ratio | a.u. | Log of ratio of sum of energy in 1st 3-axis principal component over energy of 3-axis root mean squared magnitude. |
| Zero Crossing St.Dev. | Seconds | Standard deviation of time between zero crossing of 1st 3-axis principal component. |
| Zero Crossing Average | Seconds | Mean of time between zero crossing of 1st 3-axis principal component. |
| Robust Arm-Tilt | a.u. | Log of mean square root of squared X & Z axes. |
| Kurtosis | a.u. | Kurtosis of 3-axis root mean squared magnitude. |
| Sleep Coefficient | a.u. | Sum of 3-axis max-min range, binned into 16 log-scaled bins. |
| **SCL** **Skin Conductance** | | |
| Skin Conductance Value | $\mu$Siemens | Center of linear tonic SCL value fit. |
| Skin Conductance Slope | $\mu$S/Min | Intraminute slope of SCL values. |
| **TMP** **Skin Temperature** | | |
| Skin Temperature Value | $^{\circ}$C | Mean skin temperature. |
| **ALT** **Altimeter** | | |
| Altimeter St.Dev. Norm | Hectopascals | Standard deviation of altimeter readings. |

Table 19: **Details of Natural and Sampled Event Dataset Distributions.** Summary of class distributions (as a percent of total task events) for activity and mood datasets.

| Task | Class | Natural Distribution | Sampled Distribution Train | Test |
|------|-------|---------------------|-------|------|
| **Activity** | Biking | 2.54% | 8.29% | 12.63% |
| | Elliptical | 0.02% | 1.06% | 1.50% |
| | High Intensity Training | 0.22% | 2.31% | 3.19% |
| | Strength Training | 0.24% | 1.59% | 13.03% |
| | Swimming | 1.41% | 16.23% | 13.52% |
| | Running | 6.65% | 12.94% | 9.66% |
| | Walking | 88.59% | 52.93% | 43.47% |
| | Weightlifting | 0.33% | 4.65% | 3.00% |
| **Mood** | Content | 29.71% | 22.13% | 20.62% |
| | Frustrated | 11.65% | 12.95% | 17.24% |
| | Excited | 5.52% | 11.48% | 10.23% |
| | Calm | 31.73% | 21.58% | 22.96% |
| | Stressed | 21.39% | 31.86% | 28.95% |

## H  BROADER IMPACT

Wearable sensors have been shown to positively impact health and well-being by promoting physical activity and sleep. They further offer the potential to reveal previously unknown actionable health information. Foundation models increase the potential to leverage wearable data for the above applications and hold promise to enable new insights and opportunities to improve health.

We support open science principles and the value of open data for scientific research; however, we have to balance these considerations with the privacy of the participants and protection of their health data. Although the training data could be de-identified, some of the data streams could not be fully anonymized. We recognize that the inability to share data of this kind is a limitation; however we believe that the presented results enable us to share valuable insights with the community.

Looking forward, LSM may serve as the stepping stone towards generating large-scale, realistic synthetic wearable datasets. These synthetic data could mimic real-world sensor patterns without compromising participant privacy and offer a promising resource for cross-institutional research collaboration. By facilitating data sharing in this way, we can overcome the current limitations in data availability and unlock new opportunities for collaborative insights and advancements.

Finally, as discussed in Appendix D.1, health and well-being are personal and sensitive topics and the abilities of such wearable foundation models may pose privacy concerns. It is essential to recognize that these ethical considerations apply to foundation models at large, not solely our method. Anyone seeking to implement or use our model should be mindful of these concerns. Both our specific method and foundation models, in general, should be used cautiously to avoid situations where their deployment might contribute to unethical outcomes or interpretations.

## I  CODE ACKNOWLEDGEMENTS

We build our methods upon the *Scenic* project (Dehghani et al., 2022), an open source codebase for vision tasks implemented in JAX with Flax. *Scenic* provides rich infrastructure for attention-base vision models and baselines. The project page can be found here: github.com/google-research/scenic.

