# OpenReview forum: "Scaling Wearable Foundation Models"
_ICLR.cc/2025/Conference — ICLR 2025 Poster_

### Official Review · Reviewer_Y2nv · 2024-11-02

**Soundness:** 3
**Presentation:** 3
**Contribution:** 3
**Rating:** 6
**Confidence:** 4

**Summary:**

This paper describes self-supervision training methodology (infilling of randomly-masked patches) applied to a very large data set of multi-modal time series data collected by wearable devices under natural usage.   The authors describe a comprehensive analysis of the model performance as a function of training data volume, compute resources and model complexity across a set of generative tasks (imputation, extrapolation/forecasting and interpolation).  Additionally they also evaluate performance of the resulting pre-trained encoder (and compare this with a supervised learning baseline) on a pair of discrimination tasks: binary exercise/non-exercise classification and multi-class activity classification.

**Strengths:**

## This paper represents strong contributions in the following areas:

### Originality:  This paper is the first to evaluate self-supervised learning methods on a multi-modal physiological data set at this scale.  The authors describe a novel modeling approach as well as a comprehensive description of their ablation experiments.

### Quality:  The authors present rigorous, systematic and thorough evaluation of performance on the generative tasks as a function of pre-training data volume, and model size.

### Significance: The analysis of generative tasks is comprehensive and may be of interest both within the problem domain (models for multi-modal data from wearable devices) as well as more broadly.  However, the lack of compelling evidence for model generality across a range of downstream tasks may limit the broad interest.

### Clarity:  The paper is fairly clear regarding the data source (including featurization), modeling approach, performance analysis and related ablation experiments.   Some details (discussed in the section below) are missing, but this is not likely to impact the general conclusions.

**Weaknesses:**

# Brief Summary #
This paper had some important weaknesses that are listed briefly below, with more detailed comments and some suggested steps to address these weaknesses following the list:

1. [Most concerning] Lack of variety in downstream task evaluation.  The paper evaluates only two closely-related downstream tasks (binary and multi-class activity classification) that utilize the learned representations.   Given that one of the main characteristics of ‘Foundation Models’ is that they are readily adaptable to a wide variety of downstream tasks, it is a stretch to consider the LSM model to be a foundation model rather than a tool for time series interpolation/extrapolation.
2. No analysis of model performance in a manner that is agnostic to the downstream task (e.g. using measures of unsupervised embedding quality)
3. Incomplete comparison of model performance on downstream tasks against baselines— LSM performance was compared with SL-trained model having the same ViT architecture, but not against other classifier types.
4. Absence of some details and explanation on the problem setup for downstream classification tasks.
5. The authors are missing discussion/citation of some past publications that represent closely-related research (learning general-purpose representations from multi-modal physiological time series captured by wearable devices)


# Additional Discussion Details #

## Issue 1: [Most concerning] Lack of variety in downstream task evaluation.
Other than an extensive analysis of the interpolation/imputation/extrapolation tasks (which are useful proxies), the paper evaluates performance on only two closely-related downstream tasks utilizing the LSM embeddings: binary exercise/non-exercise classification, and multi-class activity classification.  This makes it hard for the reader to have confidence that LSM can truly be considered a “foundation model”.  Per the proposed terminology set forth in Bommasani, et al., ‘On the Opportunities and Risks of Foundation Models’, in addition to being trained via self-supervision on diverse large-scale data, a FM “can be adapted to a wide range of downstream tasks”.    Moreover, there would be particular value and impact in demonstrating that the LSM embeddings are useful for ***physiologically-relevant*** downstream tasks.

### Suggestions to address Issue 1:
The data source described by the authors contains some additional metadata/labels that could be used as targets for downstream tasks, building evidence for the general-purpose utility of the LSM embeddings.   The metadata includes subject-reported age, sex, and weight (possibly BMI, which is listed in table 2a) that can be used as regression and/or classification targets.

Other targets, particularly those with some direct physiological relevance, would also provide additional evidence for general-purpose utility of the model.  I recall that Fitbit generates a ‘Cardio Fitness Score' that represents an estimate of VO2max.  If VO2max is available it could be utilized as either a regression target or a classification target (simple binning into high/low or tertiles/quartiles).  This would be a good choice of task, since there is a first-principles argument that the temporal relationship between physical effort and heart rate response *should* contain information about VO2max.

## Issue 2: No analysis of model performance in a manner that is agnostic to the downstream task
In a complement to suggestions for issue 1 above (more downstream task variety) providing some task-agnostic analysis of embedding ‘quality’ would also help the argument that LSM is in fact a general-purpose foundation model. The authors provide some t-SNE visualization of embedding distribution in Figures 8 and 9, but no accompanying quantitative analysis.

### Suggestions to address Issue 2:
Include some quantitative measure of how well-distributed the model embeddings are, and/or how well 5h data segments generated by the same individual can be distinguished from segments generated by other individuals.

Multiple measures of embedding “quality” have been reported that are useful for evaluating and comparing models in a manner that is agnostic to the downstream task (for a recent review/comparison, see Tsitsulin, *et al.* ‘Unsupervised Embedding Quality Evaluation’ https://arxiv.org/abs/2305.16562).

## Issue 3: Incomplete comparison of model performance on downstream tasks against baseline.
The authors compare the performance of their pre-trained model against a supervised learning baseline for the two downstream classification tasks.   However, this baseline (ViT with 100M parameters) may not be optimal, particularly for the relatively simple task of exercise vs. non-exercise binary classification.   It’s worth noting that based on the test set summary in Table 2b simply guessing the majority class (non-exercise) every time would produce ~65% accuracy, but the 100M-parameter ViT only achieves 70.9% with supervised learning (Table 3b).

### Suggestions to address Issue 3:
Compare performance on both downstream classification tasks using an alternate modeling approach such as random forest or regularized logistic regression (or other boilerplate ML classifiers).  Given the problem statement (classify a period of time as exercise vs. non-exercise, or by exercise type, using a time series of 27 features), it is possible that RF or LR could outperform the SL ViT.  Even if they don’t, it could give an indication of the performance gap between these ‘non-deep’ supervised ML methods and the ViT SL baseline.

## Issue 4: Absence of some details and explanation on the problem setup for downstream classification tasks.
Section 4.2 on discriminative tasks is very light on details, to the extent that it would be difficult or impossible for other researchers to reproduce the findings (using their own data and models).  The data segments are 5 hours long (so presumably only a subset of that time contains the activity/exercise of interest) but it is not clear how the activity period within the segment is labeled or how the classification is evaluated. Tables 3b and 3c report mAP as a performance metric, which makes me think that there is some time segmentation or drawing of boundary-boxes involved in the downstream task, but this is not explained in the text.  In contrast, Appendix section B.3 states “This is likely as there are significant periods of walking in the 5-hour inputs, even if the activity is labeled otherwise”, which implies that the entire 5h inputs are assigned a single class label.  This is confusing (or at least unclear) to the reader.

### Suggestions to address Issue 4:
Provide more detail on the downstream classification problem setup, accompanied by a diagram or illustration (at least in the Appendix) that illustrates visually how the model outputs are evaluated against the ground truth labels.

## Issue 5: The authors are missing discussion/citation of some past publications that represent closely-related research.

Some relevant published work that is fairly similar in scope (learning general-purpose representations from multi-modal physiological time series captured by wearable devices) exists, but these are not cited or discussed.  One clear example is  D. Spathis *et al.*, Self-supervised transfer learning of physiological representations from free-living wearable data (2020, https://arxiv.org/abs/2011.12121) which utilized pretext tasks to pre-train models for multi-modal inputs from wearables (heart rate and raw IMU), then evaluated the resulting embeddings on a variety of downstream tasks.

### Suggestions to address Issue 5:
At least include the Spathis *et al.* paper (listed above) among the citations.  Given the close match in scope, this paper should probably also be included in Table 1 despite not referring to its work as a “foundation model” (being from 2020, it predates the popularization of the “foundation model”).    It may also be worth doing an additional literature review specifically looking for pre-2021 papers that do not use the term “foundation model” but are clearly pursuing the same objective.

**Questions:**

1. In section 3.1 (data preprocessing) provide an explicit statement of which physiological metrics ("features") are calculated on-device vs, which are calculated on the stored centralized data.  This section mentions broadly that the raw sensor signals are not stored centrally or on-device (which seems to imply that all of the features are calculated on-device), but whether this is the case should be stated clearly.
2. Toward the end of section 3.1 (data preprocessing) the authors state “Within each 300-minute window, missing data between valid data points was linearly interpolated, and leading missing minutes were backfilled.”  Was the **test set** that was used for evaluating generative tasks free of these missing values that had been infilled by linear interpolation/backfilling?  If so, these time points should be excluded from the calculation of generative task performance (MAE/MSE) because they do not represent real measurements (but rather synthetic values created in data preparation using a human-chosen method).  Please comment on how this issue was handled.
3. Typo(?) on line 202.  “Kurtosis is the kurtosis of the BFP signal”.  Should (I think) be BPF.
4. Why was the data input length of 5h chosen?  What other lengths were considered, and how were these compared to ultimately land on the choice of 5h.

---

> ### Author Response · Authors · 2024-11-21
> **Responses 1-2**
>
> Dear Reviewer Y2nv,
> Thank you so much for taking the time to read our paper and to provide us with such high quality feedback. We are especially appreciative of how you have provided suggestions to address your concerns. We respond to your comments and questions below.
>
> # 1. Additional Diverse Downstream Tasks.
> **Addresses "Weakness 1"**
>
> **Response:** Thank you for thoughtful comments. We agree that evaluating LSM on additional downstream tasks would help emphasize the utility of the pretraining as well as illustrate the breadth of its applications. To this end, and based on your recommendations, we have added 2 additional discriminative tasks. They are sex classification, and age classification. A description of these tasks and the performance of LSM and our baselines can be found in **Appendix Section C.7 Additional Discriminative Tasks.** A summary of these tasks can be found below as well as the results found in the paper.
>
> We find your comment regarding VO2 max particularly intriguing, and agree that a robust embedding would be able to encode such information regarding a person’s health. Unfortunately we do not have a VO2 Max label, so this remains an interesting future exploration.
>
> **i. Biological sex classification:** Classification is a data sample from a female or male person. This is a proxy to understand if the learnt embeddings encode information that can be used to discriminate difference in biology.
>
> **ii. Binned age classification:** Classification of individuals, from a sample of their wearable data into 4 age bins. This is a proxy task to understand if the learnt embeddings encode changes in physiology correlated with aging.
>
> | **Pretrain** | **Probe/FT** | **Sex Classification Acc.** | **Sex Classification mAP** | **Age Classification Acc.** | **Age Classification mAP** |
> |---|---|---|---|---|---|
> |  | Supervised | 66.0 | 60.1 | 35.3 | 33.4 |
> | | | | | | |
> | MSN | Linear Probe | 68.0 | 61.9 | 36.4 | 29.2 |
> | DINO | Linear Probe | 66.4 | 64.6 | 31.0 | 29.9 |
> | SimCLR | Linear Probe | 67.6 | 60.3 | 37.6 | 26.3 |
> | LSM MAE | Linear Probe | 75.4 | 75.3 | 40.9 | 38.2 |
> | | | | | | |
> | MSN | Fine-tune | 68.3 | 63.8 | 43.8 | 36.6 |
> | DINO | Fine-tune | 67.9 | 60.9 | 36.9 | 34.4 |
> | SimCLR | Fine-tune | 65.2 | 57.6 | 37.3 | 33.5 |
> | LSM MAE | Fine-tune | 75.4 | 79.8 | 42.2 | 38.7 |
> | | | | | | |
> |  | Gain over Supervised | +14.1% | +32.8% | +19.5% | +15.9% |
>
>
> # 2. Evaluating Embeddings.
> **Addresses "Weakness 2"**
>
> **Response:** Thank you for bringing this point to light. We agree that a quantitative measure of our shown embeddings is useful. To this end we have added **Table 12 in Appendix C.4**. to give some numerical understanding of the effect of scaling pretrained data size. These three metrics are all based on K-Means clustering and either evaluate the clustering quality of these k clusters or compare the k clusters to the true label clusters. Similar to our plotted tSNEs in **Figure 8**. We find that the scaling data size has a subtle effect on the clustering of the embeddings.

---

> ### Author Response · Authors · 2024-11-21
> **Responses 3-4**
>
> # 3. Discriminative Performance of Classical ML Methods.
> **Addresses "Weakness 3"**
>
> **Response:** This is an interesting analysis and something we had previously overlooked. To this end we have added both Random Forest and Logistic Regression Classifier results for all of our discriminative tasks (including the 2 new tasks of sex, and binned age classification) in the paper. We find that these methods exhibit strong performance, often on par with our supervised and/or linear probe methods. We find that our fine-tuned SSL methods regularly out perform these classical ML approaches. However, the strong performance of these statistical methods may be indicative of the potential to distill LSM models to small models that may be performant on the edge. We have referenced this in the discussion of **Section 5.2**. A summary of the performance of these ML baselines is added below:
>
> | | Exercise Detection | | Activity Recognition | | Sex Classification | | Age Classification | |
> |-----------------------|:------------------:|:---------------------:|:---------------------:|:----------------------:|:------------------:|:---------------------:|:---------------------:|:----------------------:|
> |                       | Acc.              | mAP                  | Acc.                 | mAP                   | Acc.                | mAP                  | Acc.                 | mAP                   | Acc.               | mAP                  |
> | Rand. Forest         | 73.0              | 76.8                 | 56.5               | 43.1                  | 66.2                | 56.6                  | 42.3               | 35.3                  |
> | Logistic Reg.        | 72.4              | 67.3                 | 60.4                 | 39.3                 | 59.9                | 54.0                  | 28.9               | 28.5                 |
> | LSM LP              | 84.7              | 89.0                 | 49.4                 | 24.6                  | 75.4                | 75.3                  | 40.9               | 38.2                 |
> | LSM FT              | 90.3              | 97.0                 | 68.5                 | 51.4                  | 75.4                | 79.8                  | 42.2               | 38.7                 |
>
>
> # 4. Clarification of Discriminative Methods.
> **Addresses "Weakness 4"**
>
> **Response:**  Thank you for this critique. It's especially helpful for us to get “fresh eyes” on our work to point out areas of ambiguity. The entire 5 hour segment is used when evaluating discriminative task performance (this is true for activity tasks as well as the recently added tasks). Of this segment the activity events vary in duration (e.g. a bike ride may be several hours, whereas a HIIT workout may only be 30 minutes).
>
> Concretely the tasks center on detecting the presence of some activity/exercise during the 5 hour duration. We agree that this strategy may not be optimal for these tasks and that a more task-specific approach may improve performance. As you note, in **Appendix B.3** (now **C.3**), it is possible that the high false positive rate for walking is due to an inappropriately sized window. Future work may explore how to dynamically time-slice the input to best extract activity relevant features only. We have added a discussion around these points in **Section 6: Limitations and Future Work.** We have additionally added a section, in the appendix specifying the methodology of the downstream discriminative tasks in order to prevent further confusion. This can be found in **Appendix E.3 Discriminative Task Tuning and Evaluation** where we write:
>
> *“In an effort to mitigate confusion regarding the implementation of our downstream discriminative tasks we cover possible questions and implementation details. All discriminative tasks operate on 5-hour wearable sensor feeds, the same form of input used during pre-training. These windowed samples are associated with meta data including a person's biological sex, age, and any activity events occurring during the 5 hour span. Activity labels are self-reported and with the corresponding events varying in duration (e.g., a bike ride may last several hours, whereas a HIIT workout may only last 30 minutes). Generally, these events are self-reported post hoc. As activity events are self-reported it is possible that other unreported events may also exist in the 5 hour span. As we discuss in **Appendix C.3**, this may result in degraded performance.”*

---

> ### Author Response · Authors · 2024-11-21
> **Response 5**
>
> # 5. Additional Related Works
> **Addresses "Weakness 5”**
>
> **Response:** Thank you for the references. These are useful for us to provide a more informed context of our work. We have updated the related work in the revised version as follows:
>
> *Learning from Multimodal Sensor Data. A significant body of work has explored representation learning for multimodal physiological time-series data from wearable devices. Spathis et al. (2021) [1] employed pretext tasks to pre-train models on multimodal inputs such as heart rate and raw IMU signals, demonstrating their effectiveness across various downstream tasks. Saeed et al. (2021) [2] introduced a self-supervised framework specifically designed for wearable sensors, emphasizing robustness through representation learning from diverse signal types. Deldari et al. (2024) [3] proposed CrossL, a cross-modal self-supervised learning approach that utilizes latent masking to effectively model interactions between modalities. Haresamudram et al. (2021) [4] applied contrastive predictive coding to human activity recognition, showcasing its capability to capture temporal dependencies in wearable sensor data. Further advancements include multitask learning for multi-dimensional clinical time series [5] and physiological measurements [6]. In contrast to prior work, our work emphasizes \textit{scalable} modeling across a broader range of wearable sensor modalities and a significantly larger data sample size. We systematically investigate scaling behavior across compute, data size, and model capacity, and examine the generalizability of the learned representations on large-scale real-world datasets.*
>
> References:
> [1] Spathis, Dimitris, et al. "Self-supervised transfer learning of physiological representations from free-living wearable data." Proceedings of the Conference on Health, Inference, and Learning. 2021.
> [2] Saeed, Aaqib, Victor Ungureanu, and Beat Gfeller. "Sense and learn: Self-supervision for omnipresent sensors." Machine Learning with Applications 6 (2021): 100152.
> [3] Deldari, Shohreh, et al. "Crossl: Cross-modal self-supervised learning for time-series through latent masking." Proceedings of the 17th ACM International Conference on Web Search and Data Mining. 2024.
> [4] Haresamudram, Harish, Irfan Essa, and Thomas Plötz. "Contrastive predictive coding for human activity recognition." Proceedings of the ACM on Interactive, Mobile, Wearable and Ubiquitous Technologies 5.2 (2021): 1-26.
> [5] Raghu, Aniruddh, et al. "Sequential multi-dimensional self-supervised learning for clinical time series." International Conference on Machine Learning. PMLR, 2023.
> [6] Narayanswamy, Girish, et al. "Bigsmall: Efficient multi-task learning for disparate spatial and temporal physiological measurements." Proceedings of the IEEE/CVF Winter Conference on Applications of Computer Vision. 2024.
>
> We have integrated the above discussion and references into the revised paper in **Appendix B** due to space limits.

---

> ### Author Response · Authors · 2024-11-21
> **Question Responses**
>
> # Question Responses
>
> ## Question 1. Feature Preprocessing.
> **Response:** We appreciate the opportunity to clarify and make our paper less ambiguous. All metrics discussed are calculated on-device. This is a practical consideration to 1. mitigate the memory and battery constraints of saving raw data, and 2. account for the bottleneck of streaming data off-device to the server side. All features are calculated with a minute granularity. The exception is heart rate which is calculated at a second granularity. We aggregate heart rate to a per-minute granularity to match the other features. We’ve added a description of this to Appendix G Additional Details of Dataset. We’ve also added clarification in **Section 3.1 Sensor Data and Preprocessing.**
>
> ## Question 2. Handling Imputed Data.
> **Response:** We thank you for giving us the chance to expound upon this detail. Much like the train set, the test set was similarly interpolated for missing values. The reported performance metrics do not account for this back filling. Although this is a limitation, which we will add to **Section 6: Limitations and Future Work**, we do not believe this invalidates our findings. LSM significantly out performs the linear interpolation baseline (the most performant baseline), which indicates that LSM is able to learn a more sophisticated representation irrespective of back propagation on imputed values.
>
> ## Question 3. Typo.
> Response: Thanks for catching this! We did mean BPF.
>
> ## Question 4. Choosing a Window Size.
> Response: We selected a 5 hour window to best balance the constraints of our pre-training and downstream tasks. Using a long window (e.g., 5 hours) during pre-training allows the model to best build an understanding of complex and long-range temporal relationships. A longer window also makes the SSL task more challenging, hopefully leading to richer representation. We want to explicitly acknowledge this as an assumption and a limitation.
>
> Regarding downstream tasks, a 5 hour window enables our generative tasks where we impute up-to 2 hours of data. For discriminative tasks, like activity recognition, different activities span varying amounts of time. We find that a 5-hour window is suitable for encompassing almost the entirety of activities. We concede that a 5 hour window may not be appropriate for all events. For example 5 hours may be too short to properly gauge a person’s sleep and may be much longer than a short walk. We have added our assumptions regarding a 5 hour window to **Section 6: Limitations and Future Work.**
>
> Looking forward, we are interested in drastically increasing the pre-training context window (using days or weeks of data) to learn longer range dependencies (e.g., how a person's sleep affects their activities, and how a person's measured data changes depending on the day of the week).

---

> ### Author Response · Authors · 2024-11-25
> **New Discriminative Task (Mood)**
>
> Thanks again for your commitment to helping us strengthen this work. In an continued effort to diversify our downstream tasks we have added the discriminative task of subject dependent mood recognition. We find that due to the innate ambiguity and personal nature of self-reported mood labels, that a user stratified method fails to learn effectively. As such our approach takes self-reported mood events, from an individual, and splits them across train and test splits. Note that the same event is not found in both splits. Also note that subject dependent mood recognition remains a challenging task due to the innate ambiguity of emotions and the range of their presented severities. The task and the corresponding results are detailed in **Appendix C.7 and in Table 14.** We have updated the discriminative task summary table below with the mood results and the classical ML baselines as well.
>
> | **Pretrain** | **Probe/FT** | **Sex Classification Acc.** | **Sex Classification mAP** | **Age Classification Acc.** | **Age Classification mAP** | **Mood Recognition Acc.** | **Mood Recognition mAP** |
> |---|---|---|---|---|---|---|---|
> | | Random Forest |66.2 | 56.6  | 42.3 | 35.3 | 41.8 | 44.4 |
> | | Logistic Regression | 59.9 | 54.0 | 28.9 | 28.5 | 43.5	| 42.8 |
> |  | Supervised | 66.0 | 60.1 | 35.3 | 33.4 | 42.4 | 32.7 |
> | | | | | | | | |
> | MSN | Linear Probe | 68.0 | 61.9 | 36.4 | 29.2 | 36.7 | 34.8 |
> | DINO | Linear Probe | 66.4 | 64.6 | 31.0 | 29.9 | 40.6 | 36.5 |
> | SimCLR | Linear Probe | 67.6 | 60.3 | 37.6 | 26.3 | 36.2 | 30.1 |
> | LSM MAE | Linear Probe | 75.4 | 75.3 | 40.9 | 38.2 | 42.0 | 34.7 |
> | | | | | | |
> | MSN | Fine-tune | 68.3 | 63.8 | 43.8 | 36.6 | 41.2 | 38.8 |
> | DINO | Fine-tune | 67.9 | 60.9 | 36.9 | 34.4 | 43.5 | 37.9 |
> | SimCLR | Fine-tune | 65.2 | 57.6 | 37.3 | 33.5 | 43.8 | 37.4 |
> | LSM MAE | Fine-tune | 75.4 | 79.8 | 42.2 | 38.7 | 49.0 | 43.0 |
> | | | | | | |
> |  | Gain over Supervised | +14.1% | +32.8% | +19.5% | +15.9% | +15.6% | 31.5% |

---

> ### Author Response · Authors · 2024-11-26
>
> Dear Reviewer Y2nv,
>
> Thanks again for investing the time and energy needed to give us such thorough feedback. We’ve recently responded to your comments and questions and have incorporated your feedback into the revised manuscript.
>
> We would greatly appreciate you taking a moment to review our responses, share your thoughts, and update your rating. We sincerely believe that your reviews have helped us strengthen our work, and we look forward to hearing from you.

---

> > ### Author Response · Authors · 2024-11-29
> > **A Gentle Reminder**
> >
> > Dear Reviewer Y2nv,
> >
> > Just a gentle reminder. We would appreciate it if you could take a moment to review our responses, share your thoughts, and update your review accordingly. Thanks again for your time, commitment, and feedback, all of which have helped strengthen our work.
> >
> > Best,
> > Authors

---

> > ### Comment · Reviewer_Y2nv · 2024-12-03
> >
> > I thank the authors for the modifications and updates to their paper, for responding to my questions concerns (as well as the concerns/questions provided by the other reviewers), and for their thorough and comprehensive summary of the experiments, analysis and edits that were done to address each point.
> >
> > I've re-read the paper with a focus on the new material, and they have addressed the majority of my concerns. While I would prefer to see a broader variety of downstream tasks evaluated for a true 'foundation model' the tasks used by the authors introduce enough variety to provide a compelling argument for generality.
> >
> > Regardless, with the author's effective edits and added material the paper is definitely stronger now.   I've increased my overall rating in response to those improvements.

---

### Official Review · Reviewer_58g1 · 2024-11-03

**Soundness:** 4
**Presentation:** 4
**Contribution:** 4
**Rating:** 8
**Confidence:** 5

**Summary:**

The authors have introduced a multimodal foundation model for sensor data. The input is mix of raw sensor samples and some preprocessed discrete derived samples due to some understandable practical constraints. They have set up a frame work to prove generative and inferential capabilities allowing the model to be regarded as foundational. With a good representative choice of model architecture, scales of data and a diverse set of scaling experiments the authors have derived a first proof of power scaling laws showing validation of feasibility for the concept of sensor foundation models.

**Strengths:**

An original research work focused on a multimodal model with true large scale of data and model size that is comparable to language foundation models. The scaling laws validate proof of principle for this area. Sample applications chosen have practical significance.

The article is demonstrating high level of original research due to the scaling law contributions. They also gain additional validation by doing so with multi modal sensor streams. There have been other notable work showing the power of pre training on single or multiple sensor streams in the past at sufficient scale but nothing addressing the scaling question.

The results and analysis were presented with sufficient clarity. Some room for improvement has been suggested in the following section.

**Weaknesses:**

While the work is generally comprehensive, some areas of improvement remain.

The limited choice of downstream tasks raises questions on the true foundational nature of the model and its few shot capability. It seems important to understand the ability to generalize to tasks beyond classification eg. regression tasks or system identification tasks.

The choice of generative evaluation goals while interesting fails to address the practical utility of the performance shown, especially with large errors in imputation. Also, there exist sophisticated imputation and interpolation schemes that are more performant comparators eg. MICE which might be the real performance benchmarks for the application.

The definition of some of the features and preprocessing is incomplete and not in the language of mathematics. Some descriptions are highly redundant eg “Kurtosis is kurtosis”. Recommend use of the language of accurate mathematical description to define the features clearly.

**Questions:**

There is a lot of work that directly deal with accelerometer data in scale and recently also PPG data. The claim that these streams are hard to interpret and therefore not in scope - doesn’t seem a valid explanation. I do understand practical constraints in collecting these streams at a high enough sampling rate and so should suffice to explain the use of intermediates such as steps or heart rate.

The graphs with the scaling laws with the marker size and color simultaneously varying and with different y-axis scales is confusing. In fact, some marker size differences are small enough to not even stand out while accounting for a significant scale increase. Request to use different marker types instead of just size and make it more obvious, and also maintain y axis scale to compare more effectively.

---

> ### Author Response · Authors · 2024-11-21
>
> Dear Reviewer 58g1,
> We really appreciate you taking the time to read our paper and provide us with such high quality feedback. Thank you for acknowledging the value of our work especially as it pertains to the exploration of scaling in the wearable domain and the potential downstream applications. We respond to your comments and questions below:
>
> # 1. Additional Downstream Tasks.
> **Addresses "Weakness 1"**
>
> **Response:** Thanks for your comment. We have added two additional tasks: 1) Biological Sex Classification; and 2) Binned Age Classification. Please refer to **Appendix C.7** for details. A summary of these tasks can be found below as well as the results.
>
> **i. Biological sex classification:** Classification is a data sample from a female or male person. This is a proxy to understand if the learnt embeddings encode information that can be used to discriminate difference in biology.
>
> **ii. Binned age classification:** Classification of individuals, from a sample of their wearable data into 4 age bins. This is a proxy task to understand if the learnt embeddings encode changes in physiology correlated with aging.
>
> | **Pretrain** | **Probe/FT** | **Sex Classification Acc.** | **Sex Classification mAP** | **Age Classification Acc.** | **Age Classification mAP** |
> |---|---|---|---|---|---|
> |  | Supervised | 66.0 | 60.1 | 35.3 | 33.4 |
> | | | | | | |
> | MSN | Linear Probe | 68.0 | 61.9 | 36.4 | 29.2 |
> | DINO | Linear Probe | 66.4 | 64.6 | 31.0 | 29.9 |
> | SimCLR | Linear Probe | 67.6 | 60.3 | 37.6 | 26.3 |
> | LSM MAE | Linear Probe | 75.4 | 75.3 | 40.9 | 38.2 |
> | | | | | | |
> | MSN | Fine-tune | 68.3 | 63.8 | 43.8 | 36.6 |
> | DINO | Fine-tune | 67.9 | 60.9 | 36.9 | 34.4 |
> | SimCLR | Fine-tune | 65.2 | 57.6 | 37.3 | 33.5 |
> | LSM MAE | Fine-tune | 75.4 | 79.8 | 42.2 | 38.7 |
> | | | | | | |
> |  | Gain over Supervised | +14.1% | +32.8% | +19.5% | +15.9% |
>
>
> # 2. More Sophisticated Generative Baselines.
> **Addresses "Weakness 2"**
>
> **Response:** Thanks for your comment, and for the suggestion of MICE[1]! We have added MICE as another baseline and updated **Table 3.a** in the main body of the paper. We show that LSM consistently outperforms MICE, likely as MICE struggles to reliably impute data in situations of structured / non-random missingness (e.g. interpolation and extrapolation). A summary of results can be found below.
>
> [1] Van Buuren, Stef, and Karin Groothuis-Oudshoorn. "mice: Multivariate imputation by chained equations in R." Journal of statistical software 45 (2011): 1-67.
>
> **Task + Method** | **10 mins** | **20 mins** | **30 mins** | **60 mins** | **120 mins**
> -------------------|--------------|--------------|--------------|--------------|---------------
> **Temporal Interpolation** | | | | |
> MICE | 0.36 / 0.42 | 0.36 / 0.43 | 0.37 / 0.43 | 0.38 / 0.45 | 0.39 / 0.48
> LSM (MAE) | **0.16** / **0.14** | **0.19** / **0.18** | **0.20** / **0.21** | **0.24** / **0.26** | **0.29** / **0.33**
> | | | | |
> **Temporal Extrapolation** | | | | |
> MICE | 0.48 / 0.66 | 0.48 / 0.66 | 0.47 / 0.65 | 0.47 / 0.65 | 0.45 / 0.64
> LSM (MAE) | **0.28** / **0.31** | **0.32** / **0.37** | **0.34** / **0.40** | **0.37** / **0.44** | **0.38** / **0.47**
> | | | | |
> **Sensor Imputation** | | | | |
> MICE | 0.30 / 0.33 | 0.30 / 0.36 | 0.31 / 0.38 | 0.33 / 0.46 | 0.37 / 0.61
> LSM (MAE) | **0.15** / **0.11** | **0.15** / **0.12** | **0.16** / **0.13** | **0.17** / **0.15** | **0.19** / **0.17**
>
>
> # 3. Improved Feature Descriptions.
> **Addresses "Weakness 3"**
>
> **Response:** We have added more detail about the computation of the features.  We appreciate the suggestion to remove redundant language and have done so.  Specifically we have edited the text in **Section 3.1** and also would like to highlight **Table 18** that has a summary of the feature definitions as well.

---

> ### Author Response · Authors · 2024-11-21
> **Question Responses**
>
> # Question Responses.
> ## Question 1. Clarification of Time Series Data Interpretability.
>
> **Response:** Thank you for giving us the chance to offer some clarification, and we apologize for any confusion. We believe you are referring to this statement *“... raw data from sensors such as accelerometers or photoplethysmography (PPG) hardware are often challenging for both consumers and experts to interpret …”.*
>
> Here, we are not attempting to explain why we use minutely data but rather to motivate the need for algorithms to convert time-series data to more human-readable forms. As you note, we use minutely features in large part due to the practical constraints of data collection.
>
> We have updated the above statement to: *“However such wearable time series data can be difficult for consumers and experts to interpret. To this end algorithms have been developed to translate time series sensor data into human-readable representations, such as step counts and heart rates.”*
>
>
> ## Question 2: Regarding Figure Styling.
>
> **Response:** Thanks for your comment. We choose different y-scales as the tasks and settings displayed are different. We tried to follow the style used in other publications to avoid confusion [1, 2]. We are experimenting with a version that uses different marker types rather than sizes and will update the paper PDF when we have a version that is clearer.
>
> [1] Xie, Zhenda, et al. "On data scaling in masked image modeling." Proceedings of the IEEE/CVF Conference on Computer Vision and Pattern Recognition. 2023.
> [2] Zhai, Xiaohua, et al. "Scaling vision transformers." Proceedings of the IEEE/CVF conference on computer vision and pattern recognition. 2022.

---

> > ### Comment · Reviewer_58g1 · 2024-11-22
> >
> > Thank you for incorporating good responses to my feedback. As such I'm satisfied with the quality of the paper and improvements presented and was already presenting a high enough rating for the work and continue to stand by it. Best of luck and I hope this work helps catalyze this area.

---

> > > ### Author Response · Authors · 2024-11-25
> > > **Thanks For Your Support and New Task**
> > >
> > > Dear Reviewer 58g1,
> > >
> > > Thanks again for your support and insight throughout this process. We thank you for maintaining your score!
> > >
> > > We'd like to highlight that in a continued effort to diversify our downstream tasks we have added the discriminative task of subject dependent mood recognition. We find that due to the innate ambiguity and personal nature of self-reported mood labels, that a user stratified method fails to learn effectively. As such our approach takes self-reported mood events, from an individual, and splits them across train and test splits. Note that the same event is not found in both splits. Also note that subject dependent mood recognition remains a challenging task due to the innate ambiguity of emotions and the range of their presented severities. The task and the corresponding results are detailed in **Appendix C.7 and in Table 14.** We have updated the discriminative task summary table below:
> > >
> > > | **Pretrain** | **Probe/FT** | **Sex Classification Acc.** | **Sex Classification mAP** | **Age Classification Acc.** | **Age Classification mAP** | **Mood Recognition Acc.** | **Mood Recognition mAP** |
> > > |---|---|---|---|---|---|---|---|
> > > |  | Supervised | 66.0 | 60.1 | 35.3 | 33.4 | 42.4 | 32.7 |
> > > | | | | | | | | |
> > > | MSN | Linear Probe | 68.0 | 61.9 | 36.4 | 29.2 | 36.7 | 34.8 |
> > > | DINO | Linear Probe | 66.4 | 64.6 | 31.0 | 29.9 | 40.6 | 36.5 |
> > > | SimCLR | Linear Probe | 67.6 | 60.3 | 37.6 | 26.3 | 36.2 | 30.1 |
> > > | LSM MAE | Linear Probe | 75.4 | 75.3 | 40.9 | 38.2 | 42.0 | 34.7 |
> > > | | | | | | |
> > > | MSN | Fine-tune | 68.3 | 63.8 | 43.8 | 36.6 | 41.2 | 38.8 |
> > > | DINO | Fine-tune | 67.9 | 60.9 | 36.9 | 34.4 | 43.5 | 37.9 |
> > > | SimCLR | Fine-tune | 65.2 | 57.6 | 37.3 | 33.5 | 43.8 | 37.4 |
> > > | LSM MAE | Fine-tune | 75.4 | 79.8 | 42.2 | 38.7 | 49.0 | 43.0 |
> > > | | | | | | |
> > > |  | Gain over Supervised | +14.1% | +32.8% | +19.5% | +15.9% | +15.6% | 31.5% |

---

### Official Review · Reviewer_9Vkn · 2024-11-04

**Soundness:** 3
**Presentation:** 4
**Contribution:** 3
**Rating:** 6
**Confidence:** 4

**Summary:**

This study scales a multimodal foundation model, LSM, across compute, data, and model size. Using a dataset of 40 million hours from 165,000 individuals, covering heart rate, heart rate variability, electrodermal activity, accelerometer, skin temperature, and altimeter data. The paper built a sensor foundation model and investigated strategies for training such models with considering data size, computation cost and model size.
The data processing steps are clearly explained and the the whole framework is tested on a variaty of tasks including generative and classification types of tasks.
The authors’ consideration of data processing and task diversity is a valuable contribution to the wearable and ubiquitous computing community, positioning this work as a significant step forward in multimodal sensor modelling. However, there are important areas for improvement, particularly regarding challenges specific to wearable computing and comparisons with state-of-the-art (SOTA) techniques.

**Strengths:**

- The data processing and model training steps are clearly outlined, and the study systematically scales the model across key factors such as data volume, computational resources, and model size.
- The paper includes a variety of tasks that effectively demonstrate the model’s utility across different contexts, enhancing the applicability of the LSM model in both generative and classification domains.

**Weaknesses:**

**Areas for Improvement**

- Background and Related Work: The paper is thin in covering prior works in multimodal wearable models. The chosen model (ViT) and baselines focus on non-wearable modalities, overlooking well-established approaches tailored to multimodal sensor data such as (but not limited to):

> [1] Saeed, A., Ungureanu, V. and Gfeller, B., 2021. Sense and learn: Self-supervision for omnipresent sensors. Machine Learning with Applications, 6, p.100152.

> [2] Deldari, S., Spathis, D., Malekzadeh, M., Kawsar, F., Salim, F.D. and Mathur, A., 2024, March. Crossl: Cross-modal self-supervised learning for time-series through latent masking. In Proceedings of the 17th ACM International Conference on Web Search and Data Mining.

> [3] Haresamudram, H., Essa, I. and Plötz, T., 2021. Contrastive predictive coding for human activity recognition. Proceedings of the ACM on Interactive, Mobile, Wearable and Ubiquitous Technologies, 5(2), pp.1-26.


- Dataset Clarity and Statistical Insights: The paper does not clearly specify the dataset utilized. The statement, “In this paper, we present the results of our scaling experiments on the largest and the most diverse wearable dataset published to date…” implies that the dataset is previously published, yet it lacks explicit details. The authors should clarify the dataset's origin and provide comprehensive information regarding its attributes. Clarification on the dataset’s availability and publication status is necessary.
Detailed statistical insights, including class distributions and sampling strategies, would enhance understanding of the dataset’s structure. This information would be particularly useful for evaluating how well the model might generalize across various activities and sensor modalities.

- Data Sampling and Task Appropriateness: The paper lacks clarity regarding the time window chosen for training and inference (e.g., It is infered that 5-hour windows is used for both generative and classification tasks). This is a critical point, as activities typically span only a few minutes, making it unclear whether such extended windows are suitable for classification. Details on data sampling strategies and the selection of window sizes should be provided to better assess model performance across tasks.

- Specific Challenges in Ubiquitous Computing: As a foundation model for wearable sensing, LSM should ideally address unique challenges in ubiquitous computing. For example:
  - the authors could discuss handling missing data at various stages (pre-training, fine-tuning, and inference), as highlighted by [2]. This would enhance the model’s robustness for real-world applications.
  - To demonstrate the model’s generalizability, it would be beneficial to extend experiments to include additional datasets covering diverse tasks, devices, and activities to analyse adaptability to a wider range of scenarios in wearable sensing.

**Questions:**

As mentioned above.

---

> ### Author Response · Authors · 2024-11-21
> **Response 1**
>
> Dear Review 9Vkn,
> We appreciate you spending the time and energy reviewing our paper, and we’re glad you found utility in our pretraining and downstream task experiments. We have responded to your comments and questions below.
>
>
> # 1. Background and Related Work.
> **Addresses "Weakness 1"**
>
> **Response:** Thank you for the references. These are useful for us in providing a more informed context of our work. We have updated the related work in the revised version as follows:
>
> *Learning from Multimodal Sensor Data. A significant body of work has explored representation learning for multimodal physiological time-series data from wearable devices. Spathis et al. (2021) [1] employed pretext tasks to pre-train models on multimodal inputs such as heart rate and raw IMU signals, demonstrating their effectiveness across various downstream tasks. Saeed et al. (2021) [2] introduced a self-supervised framework specifically designed for wearable sensors, emphasizing robustness through representation learning from diverse signal types. Deldari et al. (2024) [3] proposed CrossL, a cross-modal self-supervised learning approach that utilizes latent masking to effectively model interactions between modalities. Haresamudram et al. (2021) [4] applied contrastive predictive coding to human activity recognition, showcasing its capability to capture temporal dependencies in wearable sensor data. Further advancements include multitask learning for multi-dimensional clinical time series [5] and physiological measurements [6]. In contrast to prior work, our work emphasizes \textit{scalable} modeling across a broader range of wearable sensor modalities and a significantly larger data sample size. We systematically investigate scaling behavior across compute, data size, and model capacity, and examine the generalizability of the learned representations on large-scale real-world datasets.*
>
> References:
> [1] Spathis, Dimitris, et al. "Self-supervised transfer learning of physiological representations from free-living wearable data." Proceedings of the Conference on Health, Inference, and Learning. 2021.
> [2] Saeed, Aaqib, Victor Ungureanu, and Beat Gfeller. "Sense and learn: Self-supervision for omnipresent sensors." Machine Learning with Applications 6 (2021): 100152.
> [3] Deldari, Shohreh, et al. "Crossl: Cross-modal self-supervised learning for time-series through latent masking." Proceedings of the 17th ACM International Conference on Web Search and Data Mining. 2024.
> [4] Haresamudram, Harish, Irfan Essa, and Thomas Plötz. "Contrastive predictive coding for human activity recognition." Proceedings of the ACM on Interactive, Mobile, Wearable and Ubiquitous Technologies 5.2 (2021): 1-26.
> [5] Raghu, Aniruddh, et al. "Sequential multi-dimensional self-supervised learning for clinical time series." International Conference on Machine Learning. PMLR, 2023.
> [6] Narayanswamy, Girish, et al. "Bigsmall: Efficient multi-task learning for disparate spatial and temporal physiological measurements." Proceedings of the IEEE/CVF Winter Conference on Applications of Computer Vision. 2024.
>
> We would like to also acknowledge that our paper builds upon a large body of existing work regarding the modeling of multimodal wearable data. The focus of our work was to explore if scaling laws, as seen in other domains, applied to wearable sensor data. To this end we explored methodologies which had proven scaling potential (e.g. ViT backbone models). Similarly, we chose baselines using well known, general-purpose SSL methodologies (e.g., reconstruction, contrastive, self-distillation) in order to inform the reader how general classes of self-supervised learning may affect the usefulness of the learned wearable representations. Our future work will focus on exploring more specific pre-training methods and architectures for wearable data. We believe that benchmarking against domain specific methods will best highlight future architecture and training methodology contributions.
>
> We have integrated the above discussion and references into the revised paper in **Appendix B** due to space limits.

---

> ### Author Response · Authors · 2024-11-21
> **Responses 2-3**
>
> # 2. Dataset Source and Potential to Open Source.
> **Addresses "Weakness 2"**
>
> **Response:** Thank you for giving us the chance to offer some clarification. This dataset has not been previously published, and was collected in-house. We are adamant supporters of open science principles, and we value open data for research. However, we have had to balance these principles with the privacy considerations of our participants and thus are unable to open source the dataset. We recognize that this is a limitation, but we felt that we needed to give the participants strong reassurance of their privacy. We are currently exploring the feasibility of open-sourcing model weights to trusted clients. We acknowledge this limitation in **Appendix H: Broader Impact.**
>
> Additionally, we agree that more information regarding the dataset and sampling would be of use. We have added dataset label distributions in **Tables 2 and Table 13**. We have additionally added a table to **Table 19 to Appendix G** that describes the natural distribution of event classes from which we sample more balanced downstream task sets. We have additionally clarified that 5-hour windows were randomly sampled for our pretraining set (and thus are not biased towards specific events) in **Section 3.2 Building a Large Scale Pretraining Sensor Dataset.**
>
>
> # 3. Data Sampling and Task Appropriateness.
> **Addresses "Weakness 3"**
>
> Response: Thank you for your thoughtful comments regarding the choice of the 5-hour time window for training and inference and its suitability for different tasks. We agree that the choice of window length is critical and may need to differ between pretraining and downstream tasks. In our study, we selected a 5-hour window as a compromise to balance the requirements of pretraining and downstream applications. Larger windows during pre-training allow the model to capture more complex and long-range temporal relationships, making the self-supervised tasks more challenging and potentially leading to richer representations. For downstream tasks, however, shorter windows may indeed be more appropriate, especially for tasks like activity recognition, where events typically last only a few minutes. It should be noted that activities significantly in duration (e.g. a marathon may last several hours), and a 5 hour window ensures that we are able to capture almost all activities in their entirety.
>
> We acknowledge that a more task-specific approach could further optimize performance. For instance, future work could create datasets with tailored window sizes for downstream tasks, highlighting event-specific time periods while masking unrelated regions. Additionally, masking strategies during downstream classification evaluation could be refined to emphasize task-relevant time spans, such as periods when certain activities occur.
>
> We have added more discussion around these points in **Section 6: Limitations and Future Work.**

---

> ### Author Response · Authors · 2024-11-21
> **Response 4**
>
> # 4. Challenges in UbiComp and Diverse Downstream Tasks.
> **Addresses "Weakness 4"**
>
> **Response:** We agree that addressing missing data is a critical aspect of real-world wearable sensing. Our model inherently handles missingness during pretraining by leveraging random masking strategies, which mimics the sporadic and incomplete nature of sensor data in real-world applications. This approach equips the model to infer missing values robustly across different stages of training and inference. While our current study focused on this masking-based strategy, we acknowledge that a deeper investigation into principled methods for handling missing data (e.g., missing not at random scenarios) is essential for further enhancing robustness. We have added this as a limitation in the paper (Section 6) and will address it in future work.
>
> We think future work may also explore the performance of downstream classification tasks in the presence of missing or incomplete data. An investigation into a model’s tolerance for “missingness”, or its ability to impute and then perform downstream tasks are both interesting questions we plan to address in future work. We have added a discussion of this to **Appendix D.**
>
> Additionally, as prompted by reviewer ZN2N, we have added discussion regarding how to scale LSM like models to the edge under **Appendix D.1.** Understanding how these large models scale to the edge is critical to maximizing their usefulness and usability, and we believe such a discussion also addresses many challenges in ubiquitous computing.
>
> We recognize that different devices offer varying arrays of sensors, and that extending our experiments to diverse devices would provide valuable insights into the model's adaptability. However, this is beyond the scope of the current study. That said, to strengthen our claims regarding downstream performance, we have included experiments on three additional tasks, which demonstrate the model’s ability to generalize effectively within the scope of our dataset.
>
> Specifically we add the tasks of sex classification, and age classification to **Appendix Section C.7 Additional Discriminative Tasks.** A summary of these tasks can be found below as well as the results.
>
> **i. Biological sex classification:** Classification is a data sample from a female or male person. This is a proxy to understand if the learnt embeddings encode information that can be used to discriminate difference in biology.
>
> **ii. Binned age classification:** Classification of individuals, from a sample of their wearable data into 4 age bins. This is a proxy task to understand if the learnt embeddings encode changes in physiology correlated with aging.
>
> | **Pretrain** | **Probe/FT** | **Sex Classification Acc.** | **Sex Classification mAP** | **Age Classification Acc.** | **Age Classification mAP** |
> |---|---|---|---|---|---|
> |  | Supervised | 66.0 | 60.1 | 35.3 | 33.4 |
> | | | | | | |
> | MSN | Linear Probe | 68.0 | 61.9 | 36.4 | 29.2 |
> | DINO | Linear Probe | 66.4 | 64.6 | 31.0 | 29.9 |
> | SimCLR | Linear Probe | 67.6 | 60.3 | 37.6 | 26.3 |
> | LSM MAE | Linear Probe | 75.4 | 75.3 | 40.9 | 38.2 |
> | | | | | | |
> | MSN | Fine-tune | 68.3 | 63.8 | 43.8 | 36.6 |
> | DINO | Fine-tune | 67.9 | 60.9 | 36.9 | 34.4 |
> | SimCLR | Fine-tune | 65.2 | 57.6 | 37.3 | 33.5 |
> | LSM MAE | Fine-tune | 75.4 | 79.8 | 42.2 | 38.7 |
> | | | | | | |
> |  | Gain over Supervised | +14.1% | +32.8% | +19.5% | +15.9% |

---

> > ### Author Response · Authors · 2024-11-25
> > **Added Mood Task**
> >
> > Thanks again for your commitment to helping us strengthen this work. In an continued effort to diversify our downstream tasks we have added the discriminative task of subject dependent mood recognition. We find that due to the innate ambiguity and personal nature of self-reported mood labels, that a user stratified method fails to learn effectively. As such our approach takes self-reported mood events, from an individual, and splits them across train and test splits. Note that the same event is not found in both splits. Also note that subject dependent mood recognition remains a challenging task due to the innate ambiguity of emotions and the range of their presented severities. The task and the corresponding results are detailed in **Appendix C.7 and in Table 14.** We have updated the discriminative task summary table below:
> >
> > | **Pretrain** | **Probe/FT** | **Sex Classification Acc.** | **Sex Classification mAP** | **Age Classification Acc.** | **Age Classification mAP** | **Mood Recognition Acc.** | **Mood Recognition mAP** |
> > |---|---|---|---|---|---|---|---|
> > |  | Supervised | 66.0 | 60.1 | 35.3 | 33.4 | 42.4 | 32.7 |
> > | | | | | | | | |
> > | MSN | Linear Probe | 68.0 | 61.9 | 36.4 | 29.2 | 36.7 | 34.8 |
> > | DINO | Linear Probe | 66.4 | 64.6 | 31.0 | 29.9 | 40.6 | 36.5 |
> > | SimCLR | Linear Probe | 67.6 | 60.3 | 37.6 | 26.3 | 36.2 | 30.1 |
> > | LSM MAE | Linear Probe | 75.4 | 75.3 | 40.9 | 38.2 | 42.0 | 34.7 |
> > | | | | | | |
> > | MSN | Fine-tune | 68.3 | 63.8 | 43.8 | 36.6 | 41.2 | 38.8 |
> > | DINO | Fine-tune | 67.9 | 60.9 | 36.9 | 34.4 | 43.5 | 37.9 |
> > | SimCLR | Fine-tune | 65.2 | 57.6 | 37.3 | 33.5 | 43.8 | 37.4 |
> > | LSM MAE | Fine-tune | 75.4 | 79.8 | 42.2 | 38.7 | 49.0 | 43.0 |
> > | | | | | | |
> > |  | Gain over Supervised | +14.1% | +32.8% | +19.5% | +15.9% | +15.6% | 31.5% |

---

> ### Author Response · Authors · 2024-11-26
>
> Dear Reviewer 9Vkn,
>
> Thanks again for investing the time and energy needed to give us such thorough feedback. We’ve recently responded to your comments and questions and have incorporated your feedback into the revised manuscript.
>
> We would greatly appreciate you taking a moment to review our responses, share your thoughts, and update your rating. We sincerely believe that your reviews have helped us strengthen our work, and we look forward to hearing from you.

---

> > ### Author Response · Authors · 2024-11-29
> > **A Gentle Reminder**
> >
> > Dear Reviewer 9Vkn,
> >
> > Just a gentle reminder. We would appreciate it if you could take a moment to review our responses, share your thoughts, and update your review accordingly. Thanks again for your time, commitment, and feedback, all of which have helped strengthen our work.
> >
> > Best,
> > Authors

---

### Official Review · Reviewer_47Qy · 2024-11-04

**Soundness:** 3
**Presentation:** 3
**Contribution:** 4
**Rating:** 8
**Confidence:** 3

**Summary:**

This paper presents a foundational model training for daily activity recognition from wrist worn portable devices. It uses a multimodal approach including 26 features out of four modalities (Acc, PPG, skin Temp  and conductance).
The model presents an improved performance compared to other methods in classification tasks of activities after user-based self-labelled classes (mostly on sports activities), as well as in interpolation and imputation tasks.
The paper analyzes the model efficiency, and complexity needed by comparing on number of subjects, and samples needed.

**Strengths:**

1. Several tasks have been defined for multimodal model evaluation: imputation, interpolation
2. A robust set of activities and extensive dataset for training 165k people.
3. Comparative analysis across model parameters and data size through an ablation study.
4. Analysis of multiple tasks: Classification, interpolation, reputation and extrapolation.

**Weaknesses:**

1. 16 features extracted from PPG, Acceleration, Skin temperature and conductance, and altimetry - seems a very reduced space for learning.
2. Temporal interpolation at the scale of 1 minute is not a very accurate task for wearable human data, as it only holds under strong assumptions of human behaviour. E.g. continuous activity, unchanged environment, no external inputs, among others.
3. It needed to be clarified the number of individuals used for training and the methods for data labelling.
4. Baseline method comparison for imputation, interpolation and extrapolation was rather simple and did not included other advance methods, see 1.
[1] Maksims Kazijevs, Manar D. Samad. Deep imputation of missing values in time series health data: A review with benchmarking. (2023) Journal of Biomedical Informatics
5. The title is too broad and uninformative --> this model is trained for sports activity and proof given in that space only of the wrist wearable data.

**Questions:**

1. How was decided the feature space?
2. How many individual training samples per person were used?
3. Did you use user-stratified splits? Or what is the data split method?
4. Why are the baseline methods for interpolation and imputation only Mean, NN and Linear?

**Details Of Ethics Concerns:**

The paper claims new data of 165k individuals, which should be verified to be approved by an ethical review board.
The data anonymization as well should be checked.
Distribution of the population should report the limited applicability to what type of individuals - inclusion/exclusion criteria in this protocol.

---

> ### Author Response · Authors · 2024-11-21
> **Responses 1-3**
>
> Dear Reviewer 47Qy,
> Thank you for taking the time to read and review our work! We appreciate your acknowledgement of the value of exploring scaling in the wearable sensor domain. We have responded to your comments and concerns below.
>
>
> # 1. The Feature Space
> **Addresses "Weakness 1" and “Question 1”.**
>
> **Response:** Thanks for giving us the opportunity to clarify. There were 26 features extracted from the PPG, accelerometer, skin temperature and skin conductance. We realize this is a relatively small amount of input features and this was part of the rationale for increasing the time window to 300 minutes which is a relatively long time period for short term activities (e.g., running). It is possible that additional features from the same sensors would have been redundant. In part, the use of these features is a practical constraint. Due to the bottleneck of streaming data off wearable devices these features are calculated on device and streamed in lieu of raw data signals. We reference this in Section 3.1 Sensor Data and Preprocessing.
>
>
> # 2. Why Minute Time Granularity.
> **Addresses "Weakness 2".**
>
> **Response:** Thank you for giving us the opportunity to add some clarification. We agree that minute granularity wearable data may not be optimal for all activities and life events. In large part the choice of a minute-level granularity was a practical constraint of the data collection process. We reference this in Section 3.1 Sensor Data and Preprocessing. The majority of these features are aggregated at the minute level on device. Though we concede that minute level features hold strong assumptions of human activity we find that there is some level of discriminative power at this scale.
>
> To this end we add a qualitative example of minutely wearable data across different activities aggregated across samples. This can be seen in **Figure 14 of Appendix G**. This plot indicates that there are differences (albeit subtle) for different activities at the minutely resolution. For example walking and weightlifting result in a smaller increase in heart rate as opposed to swimming. We have added some discussion of this as well in **Appendix G.**
>
>
> # 3. Understanding Dataset Statistics, Splits, and Methods.
> **Addresses "Weakness 3", “Question 2”, and “Question 3”.**
>
> **Response:** Thanks for your comment. The dataset  was  split  80-20  based  on  subjects (user-stratified)  into  train-test  splits  (132072  subjects  in  training, 33018 subjects in testing). Specifically for our pre training set we take 10 samples of wearable data per subject. The labels for downstreams tasks are self-reported, meaning the user reports their own activity events, and demographic information. In these datasets as well the dataset is split appropriately 80-20 based on subjects.
>
> We updated corresponding text at **Section 3.2 Building a Large Scale Pretraining Sensor Dataset, Section 4.2 Discriminative Tasks**, and in **Appendix C.7 Additional Discriminative Tasks.**

---

> ### Author Response · Authors · 2024-11-21
> **Responses 4-5**
>
> # 4. Generative Task Baselines.
> **Addresses "Weakness 4", and “Question 4”.**
>
> **Response:** Thanks for your comment. We agree that more advanced imputation methods may shed additional light on our system. As recommended in this review article and by reviewer 58g1 we implement MICE (Multivariate Imputation by Chained Equations) [1] and have updated **Table 3.a** in the main body of the paper.  A summary of the performance of MICE vs LSM can be found below.
>
> [1] Van Buuren, Stef, and Karin Groothuis-Oudshoorn. "mice: Multivariate imputation by chained equations in R." Journal of statistical software 45 (2011): 1-67.
>
> **Task + Method** | **10 mins** | **20 mins** | **30 mins** | **60 mins** | **120 mins**
> -------------------|--------------|--------------|--------------|--------------|---------------
> **Temporal Interpolation** | | | | |
> MICE | 0.36 / 0.42 | 0.36 / 0.43 | 0.37 / 0.43 | 0.38 / 0.45 | 0.39 / 0.48
> LSM (MAE) | **0.16** / **0.14** | **0.19** / **0.18** | **0.20** / **0.21** | **0.24** / **0.26** | **0.29** / **0.33**
> | | | | |
> **Temporal Extrapolation** | | | | |
> MICE | 0.48 / 0.66 | 0.48 / 0.66 | 0.47 / 0.65 | 0.47 / 0.65 | 0.45 / 0.64
> LSM (MAE) | **0.28** / **0.31** | **0.32** / **0.37** | **0.34** / **0.40** | **0.37** / **0.44** | **0.38** / **0.47**
> | | | | |
> **Sensor Imputation** | | | | |
> MICE | 0.30 / 0.33 | 0.30 / 0.36 | 0.31 / 0.38 | 0.33 / 0.46 | 0.37 / 0.61
> LSM (MAE) | **0.15** / **0.11** | **0.15** / **0.12** | **0.16** / **0.13** | **0.17** / **0.15** | **0.19** / **0.17**
>
>
> # 5. Clarification of Pre-Training Data Distribution.
> **Addresses "Weakness 5"**
>
> **Response:** Thank you for the feedback. We would like to clarify that the pretraining dataset is not limited to sports activity data. It encompasses a wide range of randomly sampled 5 hour windows over an extended period (January 1st, 2023 to July 2nd, 2024), and is thus composed of diverse daily-living scenarios. While we showcase downstream performance on activity-related tasks as illustrative examples, the pretraining data includes comprehensive multimodal signals beyond specific activities, supporting the model’s ability to generalize across various applications. To this end we find that our added discriminative tasks of biological sex classification, and binned age classification (as added in **Appendix Section C.7 Additional Discriminative Tasks**) help show the utility of this more general wearable pre-training.

---

> ### Author Response · Authors · 2024-11-26
>
> Dear Reviewer 47Qy,
>
> Thanks again for investing the time and energy needed to give us such thorough feedback. We’ve recently responded to your comments and questions and have incorporated your feedback into the revised manuscript.
>
> We would greatly appreciate you taking a moment to review our responses, share your thoughts, and update your rating if you find it appropriate. We sincerely believe that your reviews have helped us strengthen our work, and we look forward to hearing from you.

---

> > ### Author Response · Authors · 2024-11-29
> > **A Gentle Reminder**
> >
> > Dear Reviewer 47Qy,
> >
> > Just a gentle reminder. We would appreciate it if you could take a moment to review our responses, share your thoughts, and update your review if you find it appropriate. Thanks again for your time, commitment, and feedback, all of which have helped strengthen our work.
> >
> > Best,
> > Authors

---

### Official Review · Reviewer_ZN2N · 2024-11-06

**Soundness:** 2
**Presentation:** 3
**Contribution:** 2
**Rating:** 5
**Confidence:** 5

**Summary:**

This work proposed a foundation model aiming to serve as a general data encoder for wearable time series data. The author leverages signals capturing a variety of physiological activities for pre-training including activities of heart, skin, and motion. The model is evaluated on the tasks including time series imputation, forecasting, and human activity recognition. The author also shows that the scaling law also applied on large models for the modality of wearable signals.

**Strengths:**

Strengths:
1. The experiments are conducted on a large scale, and modeling a variety of signals including activities of heart, skin, and motion.
2. This work empirically shows that the scaling law of modeling can also be applied to the modality of wearable signals, in terms of scaling up computability, size of dataset, and size of model.
3. The work includes reasonable baseline comparison with vision-based models.

**Weaknesses:**

Weaknesses:
1. The model has fixed shape input, which raises concern about generalization. For example, it is very common that different devices and scenarios have different sets of sensors with different sampling rate. Some wearables have PPG + inertial sensor data only. Some have other more sophisticated sensors like Galvanic Skin Response and Electrocardiogram. It’s unclear exactly how such a model would work with different sets of input modalities with diverse sampling and data resolution settings.

2. It is not clear why modeling the time series wearable signal as an image only as opposed to other alternatives like spectral representations (fourier transformation, wavelet transformation). Wearable signals are essentially time series, and are more similar to audio modality. What is the intuition behind this design choice? Also, no comparison has been provided.
[I] Huang, Po-Yao, et al. "Masked autoencoders that listen." Advances in Neural Information Processing Systems 35 (2022): 28708-28720.
[II] Ansari, Abdul Fatir, et al. "Chronos: Learning the language of time series." arXiv preprint arXiv:2403.07815 (2024).

3. Insufficient description of modeling strategy and motivation with lack of model description. The paper can benefit from adding a more comprehensive comparison with general time series models like Chronos [2].

4. The key take away “the larger the model the better the performance” “the larger the dataset the better the performance” are fairly standard statements in the AI/ML area at this point. The paper can be improved by bringing in more insights with respect to the model size, scalability in different conditions (e.g., resource constraint/edge computing setting for example, or a distributed setting). Also, large offline dataset during training is hard to come by and often comes with significant privacy concerns. So when it comes to scalability, the computational resource and privacy are the main bottleneck. I think a bit more discussion around the key takeaways and the model’s scalability contributions would be important.

5. Only a single real world downstream task of activity recognition has been explored in the current version of the work. While I highly appreciate the large-scale data collection efforts (that enabled the work) and the large computational model, only a single downstream task fails to highlight its capability. The paper can benefit from incorporating other real world mobile health applications such as stress/fatigue modeling with similar wearable data or may be other physiological conditions modeling.

6. Activity recognition is a well studied area. One additional task that could be interesting is whether such a model generalizes across unseen classes (e.g., dancing, cooking, and other activities for daily living). This could further highlight the capabilities of the model.

7. One of the core strengths of the paper is its large dataset. Making it available to the broader community could help research and development in this area. I could not find any mention about open sourcing the dataset. Also, no mention about open sourcing the model either. I understand that often there are IRB restrictions but I think making the model and data available could be a game changer in this area.

8. Relatively minor comment but there are some repetitive statements. The paper can be shrunk a bit and can be made it a bit more compact.

**Questions:**

Please refer to the weaknesses for detailed description of each points.

1. How the model would generalize to additional wearable activity tracking tasks? or event wearable health tasks?
2. Would the model and the representation generalize to unknown classes?
3. Can the authors provide more comparisons with time series models?
4. Describe the significance of the key take away of the paper. How it benefits the scalability of the model/approach in different settings.
5. Can the model be use for more diverse down stream tasks?
6. Clarify if you would open source the data and model?
7. Polish the write up and compress the text a little bit.

**Details Of Ethics Concerns:**

For potential data release, the projects may go through responsible research practice checks.

---

> ### Author Response · Authors · 2024-11-21
> **Responses 1-3**
>
> Dear Reviewer ZN2N,
> Thank you for your thoughtful and thorough feedback.
> We have addressed your comments and questions below.
>
>
> # 1. Generalizing to Different Sensors and Sample Rates.
> **Addresses "Weakness 1"**
>
> **Response:** We agree that creating generic input representations for data from different sensors each with different properties is challenging. This was part of the reason we selected to build our model on minutely features derived from the raw sensors which may vary in sample rates. Similar minutely features could be derived from a range of sensors making the inputs somewhat independent of these devices. Adapting this model to other devices, with different sensor features is difficult. We want to explicitly acknowledge these points as assumptions in our discussion of limitations. These points have been added to **Section 6: Limitations and Future Work.**
>
>
> # 2. Representation of Time-Series Data.
> **Addresses "Weakness 2"**
>
> **Response:** Wearable signals in their “raw” form (e.g., 120Hz accelerometer values) often exhibit strong periodicity. However, our derived minutely input features do not always exhibit the same properties.  As a result we focused on time-domain representations as our inputs.  We have added an acknowledgement that this may not be optimal in **Section 6: Limitations and Future Work**. Additionally, as mentioned below (in response 3) We have added relevant references to prior time series models (like Chronos and MAEs that Listen) in **Section 5.1: Training Procedures**, and have additionally added a section of related work relevant for learning from time-series sensor data in the **Appendix Section B: Additional Related Work.**
>
>
> # 3. Additional Modeling Strategy Details and References and Comparisons to Other Time Series Models.
> **Addresses "Weakness 3" and “Question 3”.**
>
> **Response:** We appreciate the comment and have added greater motivation for the choices of using MAE and ViT in **Section 5.1: Training Procedures.** In summary, these modeling strategies have already been shown to effectively scale in order domains. We focus on exploring if these same scaling laws hold true for wearable data rather than comparing architectures.
>
> We agree that a discussion of general time-series models is informative, and thus have emphasized that general time series models, including Chronos, are relevant work. However, we find that these time series models (e.g. Chronos, TiDE, PatchTST) are univariate forecasters (though the input may be multivariate). For this reason it is difficult to fairly compare them against our multivariate forecasting model. We believe that adapting these general time series models to perform multivariate forecasting is an interesting line of research that likely warrants its own exploration. We have added references to Chronos in **Section 5.1: Training Procedures.**
>
> Reviewer 58g1 suggested the use of MICE (Multivariate Imputation by Chained Equations) as an additional and more sophisticated generative time-series baseline. We have added MICE [1] to **Table 3.a.** In summary the added results include:
>
> [1] Van Buuren, Stef, and Karin Groothuis-Oudshoorn. "mice: Multivariate imputation by chained equations in R." Journal of statistical software 45 (2011): 1-67.
>
> **Task + Method** | **10 mins** | **20 mins** | **30 mins** | **60 mins** | **120 mins**
> -------------------|--------------|--------------|--------------|--------------|---------------
> **Temporal Interpolation** | | | | |
> MICE | 0.36 / 0.42 | 0.36 / 0.43 | 0.37 / 0.43 | 0.38 / 0.45 | 0.39 / 0.48
> LSM (MAE) | **0.16** / **0.14** | **0.19** / **0.18** | **0.20** / **0.21** | **0.24** / **0.26** | **0.29** / **0.33**
> | | | | |
> **Temporal Extrapolation** | | | | |
> MICE | 0.48 / 0.66 | 0.48 / 0.66 | 0.47 / 0.65 | 0.47 / 0.65 | 0.45 / 0.64
> LSM (MAE) | **0.28** / **0.31** | **0.32** / **0.37** | **0.34** / **0.40** | **0.37** / **0.44** | **0.38** / **0.47**
> | | | | |
> **Sensor Imputation** | | | | |
> MICE | 0.30 / 0.33 | 0.30 / 0.36 | 0.31 / 0.38 | 0.33 / 0.46 | 0.37 / 0.61
> LSM (MAE) | **0.15** / **0.11** | **0.15** / **0.12** | **0.16** / **0.13** | **0.17** / **0.15** | **0.19** / **0.17**

---

> ### Author Response · Authors · 2024-11-21
> **Responses 4-5**
>
> # 4. Additional Discussion of Scaling and Scaling to the Edge.
> **Addresses "Weakness 4" and “Question 4”.**
>
> **Response:** Thank you for this suggestion. We agree that additional discussion on this topic adds value to the paper. We have added a discussion section titled **Can Wearable Foundation Models Scale to Edge Devices?.** Due to space constraints this has been added to **Appendix D** and referenced in **Section 5.2 Results and Discussion**.
> We write:
> *Our model saturation in 110M parameters highlights a practical advantage: models of this size can potentially run in real-time on modern mobile devices, leveraging advancements in on-device large language models that often exceed 1 billion parameters in on-device deployments [1]. We further note that the strong performance of statistical machine learning baselines, on classification tasks, indicate that there may be utility in distilling LSM to be smaller and more edge-performant. Additionally, unlike prior research focused on raw sensor data (e.g., [2]), our results establish that scaling is effective with per-minute aggregated data. This approach not only reduces privacy risks but also introduces a standardized format that is easier to unify across platforms and devices. Looking forward, these findings open the door to collaborative efforts such as federated learning [3] across different wearable manufacturers, enabling scalable, privacy-preserving training while maintaining compatibility across diverse hardware ecosystems.*
>
> [1] Jiajun  Xu,  Zhiyuan  Li,  Wei  Chen,  Qun  Wang,  Xin  Gao,  Qi  Cai,  and  Ziyuan  Ling.   On-device language models: A comprehensive review.arXiv preprint arXiv:2409.00088, 2024
> [2] Hang Yuan, Shing Chan, Andrew P Creagh, Catherine Tong, Aidan Acquah, David A Clifton, andAiden Doherty.  Self-supervised learning for human activity recognition using 700,000 person-days of wearable data.NPJ digital medicine, 7(1):91, 2024.
> [3] Wei  Yang Bryan  Lim,  Nguyen  Cong  Luong,  Dinh  Thai  Hoang,  Yutao  Jiao,  Ying-Chang  Liang,Qiang Yang, Dusit Niyato, and Chunyan Miao.  Federated learning in mobile edge networks:  A comprehensive survey.IEEE communications surveys & tutorials, 22(3):2031–2063, 2020.
>
>
> # 5. Additional Downstream Tasks.
> **Addresses "Weakness 5" and “Question 5”.**
>
> **Response:** We agree that additional downstream tasks would better highlight the capabilities of our model. To this end we have added 2 additional tasks which have been added to the appendix due to lack of space in the main paper. Results for these tasks can be seen in **Table 14** of **Appendix Section C.7 Additional Discriminative Tasks**. We have additionally added the results below.
>
> **i. Biological sex classification:** Classification is a data sample from a female or male person. This is a proxy to understand if the learnt embeddings encode information that can be used to discriminate difference in biology.
>
> **ii. Binned age classification:** Classification of individuals, from a sample of their wearable data into 4 age bins. This is a proxy task to understand if the learnt embeddings encode changes in physiology correlated with aging.
>
> | **Pretrain** | **Probe/FT** | **Sex Classification Acc.** | **Sex Classification mAP** | **Age Classification Acc.** | **Age Classification mAP** |
> |---|---|---|---|---|---|
> |  | Supervised | 66.0 | 60.1 | 35.3 | 33.4 |
> | | | | | | |
> | MSN | Linear Probe | 68.0 | 61.9 | 36.4 | 29.2 |
> | DINO | Linear Probe | 66.4 | 64.6 | 31.0 | 29.9 |
> | SimCLR | Linear Probe | 67.6 | 60.3 | 37.6 | 26.3 |
> | LSM MAE | Linear Probe | 75.4 | 75.3 | 40.9 | 38.2 |
> | | | | | | |
> | MSN | Fine-tune | 68.3 | 63.8 | 43.8 | 36.6 |
> | DINO | Fine-tune | 67.9 | 60.9 | 36.9 | 34.4 |
> | SimCLR | Fine-tune | 65.2 | 57.6 | 37.3 | 33.5 |
> | LSM MAE | Fine-tune | 75.4 | 79.8 | 42.2 | 38.7 |
> | | | | | | |
> |  | Gain over Supervised | +14.1% | +32.8% | +19.5% | +15.9% |

---

> > ### Author Response · Authors · 2024-11-25
> > **Added Task (Mood)**
> >
> > Thanks again for your commitment to helping us strengthen this work. In an continued effort to diversify our downstream tasks we have added the discriminative task of subject dependent mood recognition. We find that due to the innate ambiguity and personal nature of self-reported mood labels, that a user stratified method fails to learn effectively. As such our approach takes self-reported mood events, from an individual, and splits them across train and test splits. Note that the same event is not found in both splits. Also note that subject dependent mood recognition remains a challenging task due to the innate ambiguity of emotions and the range of their presented severities. The task and the corresponding results are detailed in **Appendix C.7 and in Table 14.** We have updated the discriminative task summary table below:
> >
> > | **Pretrain** | **Probe/FT** | **Sex Classification Acc.** | **Sex Classification mAP** | **Age Classification Acc.** | **Age Classification mAP** | **Mood Recognition Acc.** | **Mood Recognition mAP** |
> > |---|---|---|---|---|---|---|---|
> > |  | Supervised | 66.0 | 60.1 | 35.3 | 33.4 | 42.4 | 32.7 |
> > | | | | | | | | |
> > | MSN | Linear Probe | 68.0 | 61.9 | 36.4 | 29.2 | 36.7 | 34.8 |
> > | DINO | Linear Probe | 66.4 | 64.6 | 31.0 | 29.9 | 40.6 | 36.5 |
> > | SimCLR | Linear Probe | 67.6 | 60.3 | 37.6 | 26.3 | 36.2 | 30.1 |
> > | LSM MAE | Linear Probe | 75.4 | 75.3 | 40.9 | 38.2 | 42.0 | 34.7 |
> > | | | | | | |
> > | MSN | Fine-tune | 68.3 | 63.8 | 43.8 | 36.6 | 41.2 | 38.8 |
> > | DINO | Fine-tune | 67.9 | 60.9 | 36.9 | 34.4 | 43.5 | 37.9 |
> > | SimCLR | Fine-tune | 65.2 | 57.6 | 37.3 | 33.5 | 43.8 | 37.4 |
> > | LSM MAE | Fine-tune | 75.4 | 79.8 | 42.2 | 38.7 | 49.0 | 43.0 |
> > | | | | | | |
> > |  | Gain over Supervised | +14.1% | +32.8% | +19.5% | +15.9% | +15.6% | 31.5% |

---

> > > ### Comment · Reviewer_ZN2N · 2024-11-28
> > >
> > > This is an excellent addition to the paper. I appreciate you adding the mood classification task as an additional task to highlight the potential for this model's generalizability to additional tasks. Thank you! I will increase my rating for this.

---

> ### Author Response · Authors · 2024-11-21
> **Responses 6-8**
>
> # 6. Generalizing to Unseen Tasks.
> **Addresses "Weakness 6", “Question 1” and “Question 2”.**
>
> **Response:** Thank you for the comments. We believe our few-shot experiments have shown that the model generalizes to unseen activities well. These fewshot experiments can be found in **Table 11 of Appendix C.2**. To further demonstrate the generalization ability,  we have added additional downstream experiments on age, and gender classification to further illustrate the fact that the model is able to generalize well to unseen class labels - shown in **Table 14 of Appendix C.7**.
>
> We would like to note that our pre-training dataset is made up of wearable data from general daily-living,  including diverse timestamps and a wide range of events that occur in users' daily lives; and thus is not biased toward specific events, such as activities. We have added a description to **Section 3.2** to avoid this potential confusion.
>
>
> # 7. Model and Data Availability
> **Addresses "Weakness 7" and “Question 6”.**
>
> **Response:** We really appreciate this comment and we support open science principles and the value of open data for scientific research; however, it is challenging to ensure participant privacy while also making data broadly available to the academic research community. We had to balance these considerations with the privacy of the participants and protection of their health data.  There was particular concern that we would not be able to assure participants about the scope of use of their data given its sensitive nature and concerns. Although the data could be de-identified, some of the data streams could not be fully anonymized. We recognize that this is a limitation, but we felt that we needed to give the participants strong reassurance that their data would not be used for any other purpose than for the research at hand and as such gave such assurances in the informed consent process. We are exploring open-sourcing model weights to trusted clients and users, but it will take some time to formalize this commitment.
>
>
> # 8. Cleaning up Writing
> **Addresses "Weakness 8" and “Question 7”.**
>
> **Response:** Thanks for the feedback. We have found some cases of repetition and have removed these. We will continue to refine and clean the writing.

---

> > ### Comment · Reviewer_ZN2N · 2024-11-24
> > **Concerns about the viability as a wearable foundation model and ethical concerns**
> >
> > I sincerely thank the authors for their detailed response and for highlighting the high-level direction and motivation of the paper, which I find valuable. However, the paper falls short in several critical aspects, raising concerns about its viability as a foundational model and I am still leaning towards a reject decision. Let me explain my position.
> >
> > 1. By definition, a foundation model should have the ability to Handle Multiple Sensor Streams and Devices (e.g., Wearables): The paper primarily focuses on a specific device with fixed input sensor streams and uniform data collection settings (e.g., sampling rate and resolution). However, the market offers diverse wearable platforms, such as wrist-worn devices like Apple Watch and Samsung Watch, as well as wearables in other form factors, including earables, chest bands, and smart glasses. To qualify as a wearable foundation model, a framework must seamlessly function across various wearable devices.
> >
> >
> > 2. By definition, a foundation model should have the ability to solve multiple tasks with wearable signals: While activity recognition is an important field and an active area of research, at the end of the day it is just one specific task. The paper does not demonstrate any strong evidence that it can potentially generalize to other tasks such as sleep stage classification, pose estimation, digital biomarkers (just to give a few examples).
> >
> >
> > 3. By definition, a foundation model should have the ability to learn representations that will generalize across unseen/unknown classes: Human activity recognition is highly context dependent. For instance, activity classes like running or swimming are relevant for young athletes but may hold limited utility for other groups. In eldercare settings, low-impact activities such as tooth brushing and stretching are often more significant. The embedding from a foundation model should generalize to tasks with relevant but previously unseen classes. The paper does not present any evidence in support of this case.
> >
> >
> > 4. By definition, a foundation model should have the ability to generalize across multiple datasets: Even if we see/call the proposed model as a foundation model for “wearable-based human activity recognition”, it should demonstrate its performance on a broader set of physical activity classes with examples of previously unseen/unknown classes. Luckily, the physical activity recognition problem is a relatively well studied area and does have a long list of large-scale datasets with different set of physical activity classes.
> >
> > Large Scale Population Assessment of Physical Activity Using Wrist Worn Accelerometers: The UK Biobank Study
> > Doherty A, Jackson D, Hammerla N, Plötz T, Olivier P, et al. (2017) Large Scale Population Assessment of Physical Activity Using Wrist Worn Accelerometers: The UK Biobank Study. PLOS ONE 12(2): e0169649.
> >
> > Quoting from the paper “96,600 participants (93.3%) provided valid data for physical activity analyses.”
> >
> > There are several other large-scale Human Activity Recognition datasets (with data from unto ~96,600 participants) that can be used to validate the model’s performance on diverse settings with respect to the location of sensor placements, devices and activity classes [1-5]. This is not an exhaustive list of human activity recognition datasets. However, validating the model on at least a few additional datasets would strengthen its case as a credible and robust foundation model for wearable-based human activity recognition tasks.
> >
> > 5. Dataset and model sharing: The major contribution of the paper is the dataset. But in the absence of open sourcing at least part of the dataset in some form, the community will not be able to use it for benchmarking future research and development of foundation models. To the best of my understanding the model will also not be open sourced either which prevent future researchers to validate the model performance with different datasets and explore fairness of the model. Overall, I find it difficult to see how the proposed model and dataset will significantly advance the field. Neither the model, dataset, nor benchmarks exhibit the hallmarks of a foundational approach.
> >
> > 6. Ethical Concerns: In the rebuttal, the authors highlight the model's ability to address additional tasks such as age and gender classification, which raises serious ethical concerns. Should a foundation model for human activity recognition be designed to automatically detect age and gender? Were permissions obtained from IRB and ethical review boards to analyze the data and develop a model for demographic characterization? Modeling activity classes or health states is one matter, but analyzing user demographics is a very different issue with potential ethical and legal implications. I am deeply troubled by these concerns.

---

> > > ### Author Response · Authors · 2024-11-25
> > >
> > > **Response:** Thank you for your thoughtful comments and for engaging deeply with our work. We are disappointed that you felt the need to reduce your score (from a 5 to a 3) and we appreciate the opportunity to address your concerns before the end of the discussion period.
> > > ## Contextualizing Our Contributions, Phrasing, and Limitations In Prior Work
> > > First, we would like to clarify that the core contribution of our paper is not the presentation of a single wearable foundation model for all device types. Instead, our primary focus is on elucidating the **scaling laws of wearable data**, which we believe provides significant insights into this emerging research area. Furthermore, most relevant literature, such as the recent ICLR 2024 paper [1], also centers on a single wearable platform while naming their model a "foundation model." Like our work, their study did not evaluate on external datasets. We have explicitly outlined similar limitations in our paper.
> > > ## Downstream Evaluation and Understanding the Dataset
> > > Our paper demonstrates the model's generalizability across nine distinct downstream tasks, including four generative tasks (random imputation, temporal interpolation, sensor imputation, and temporal extrapolation for forecasting) and five discriminative tasks (activity recognition, exercise detection, mood classification, sex classification, and binned age classification). These results provide robust evidence that our model can handle a diverse range of tasks using wearable signals. Additionally, as highlighted in **Section 3.2**, our pretraining dataset includes daily-living activities across diverse timestamps and life events, avoiding biases toward specific events or activities. Importantly, our dataset is **not an activity-specific dataset**, as we aim for broader applicability.
> > > ## Mood Recognition (Additional Downstream Task)
> > > As mentioned above, in an effort to diversify our downstream tasks we have added the task of **subject dependent mood recognition**. Details regarding this experiment and its results can be found in **Appendix C.7 and Table 14.** We hope that this task highlights the utility of our methods outside of the space of activity recognition. We agree that additional daily living activities (e.g. brushing teeth, stretching, etc.) would be interesting and useful to explore. However, our dataset does not include such labels, making this an interesting exploration for future research.
> > > ## Label Efficient Learning and Adapting to Unknown Classes
> > > Regarding generalization to unseen classes, we demonstrate this through **few-shot learning experiments** (see **Figure 6 and Appendix C.2**), where our model shows comparatively strong performance with only 5, 10, 15, and 20 shots per class. It is also not clear how to conduct zero-shot learning experiments without integrating a language model. This is beyond the scope of this paper.
> > > ## Data Sharing and Model Sharing
> > > On the matter of dataset and model sharing, we acknowledge the importance of open-source contributions to the research community. However, it is common practice for models trained on large-scale user datasets not to be open-sourced. For instance, the ICLR 2024 paper [1] did not open-source their model weights or datasets. This is also observed in other domains, such as language models (e.g., GPT and Gemini), where sharing weights or datasets is rare. Nevertheless, we strongly believe the insights provided in our work, including detailed evaluations, significantly advance understanding in the field, and we will continue to explore the possibility of legally and ethically sharing our models.
> > > ## Addressing Ethical Concerns
> > > With respect to ethical concerns, we emphasize that age and sex classification were included as **demonstrative downstream tasks** to showcase the versatility of our model. These tasks are common in wearable research, as also seen in the ICLR 2024 paper [1], which included similar tasks. Furthermore, the inclusion of metadata-based tasks was **explicitly recommended by another reviewer** (Reviewer Y2nv). **All data used in this study were collected with full legal consent and ethical approvals for research purposes.** Importantly, we are not advocating for the practical use of these predictions, nor do we intend to leverage these results outside of a purely academic context.
> > > We hope this response addresses your concerns and clarifies the contributions of our work. Thank you again for your valuable feedback.
> > >
> > > ## Reference
> > > [1] Abbaspourazad, Salar, et al. "Large-scale Training of Foundation Models for Wearable Biosignals." The Twelfth International Conference on Learning Representations.

---

> > > > ### Author Response · Authors · 2024-11-26
> > > > **Contextualizing Large Sensor Models and Response to Concerns**
> > > >
> > > > # Contextualizing Large Sensor Models and Response to Concerns
> > > > Thank you again for your feedback. We appreciate your perspective and acknowledge that foundation models are intended to be flexible models. However, we would argue that the statement that a foundation model must, by definition, be able to handle multiple sensor streams and devices may not be entirely justified. As highlighted in Bommasani et al. [1], foundation models are primarily defined by their training on vast and diverse datasets (data scaling), their adaptability to a range of downstream tasks, and their use of large-scale architectures (model scaling). As emphasized by Bommasani et al. [1], transfer learning underpins their versatility, while scaling amplifies their power. While some foundation models exhibit multimodal capabilities, this is not a universal requirement;  for example, early GPT models focus on text, and models like DALL-E specialize in images. As such, handling multiple devices, similarly, should be viewed as an application-specific extension rather than a core defining characteristic. Our work aligns with these principles, emphasizing scaling laws and exploring multimodal (multiple sensor streams) and opening opportunities for multi-device extensions in the wearable domain.  We could potentially reduce the use of the term “foundation model” in the paper and rather simply focus on scaling properties if that would help?
> > > >
> > > >
> > > > With regards to our investigations of demographic -specific features, the ability of a model to predict demographic information such as age and gender does not necessarily imply that the model is explicitly designed to capture this information. As shown in prior work, even randomly initialized neural networks can predict demographic attributes such as age or gender in medical imaging contexts (e.g., chest X-rays) without targeted design [2]. Similarly, in our case, the observed results for predicting demographic  attributes are modest and achieved through fine-tuning specific to this task, rather than being an inherent or intentionally designed capability of the model. Once again, we would like to highlight that these were tasks that another reviewer (Y2nv), felt would strengthen this work.
> > > > We have added further discussion to capture these points in **Appendix D.1** where we discuss **“Contextualizing Wearable Foundation Models”** and **“Understanding Demographic Proxy Tasks”** and in **Appendix H: Broader Impact**. We hope that these changes address your concerns and would be happy to follow up with any further discussion if necessary.
> > > >
> > > > ## References
> > > > [1] Bommasani, Rishi, et al. "On the opportunities and risks of foundation models." arXiv preprint arXiv:2108.07258 (2021).
> > > > [2] Glocker, Ben, et al. "Algorithmic encoding of protected characteristics in chest X-ray disease detection models." EBioMedicine 89 (2023).

---

> > > > > ### Comment · Reviewer_ZN2N · 2024-11-28
> > > > >
> > > > > Thank you for the response.
> > > > >
> > > > > I generally agree with your sentiments and points here. Wearable foundation model is a relatively new and emerging field. The field is not there yet and a lot of research is needed before we could see a greater form of generalizability and flexibility (e.g., across multiple and diverse tasks, devices, sensor streams, datasets, output classes). but since the paper is being presented as a "Foundation model" paper, I was looking to see at least one or two of these characteristics in the paper. There comes my dilemma.
> > > > >
> > > > > While I still don't appreciate the age and gender classification task, I want to give you credit for adding the mood classification task. I also agree with your suggestion "We could potentially reduce the use of the term “foundation model” in the paper and rather simply focus on scaling properties if that would help?".
> > > > >
> > > > > I will update my score reflecting these changes/improvements.

---

> > > > ### Comment · Reviewer_ZN2N · 2024-11-26
> > > >
> > > > Thank you for the inclusion of the mood classification task. Overall, my main concerns are (1) generalizability of the proposed model to diverse tasks, (2) generalizability to different sensing modality sets, and (3) ethical concerns around sex and age modeling. The mood classification task partially addressed the first concern. The paper would significantly improve if you could include at least one or two more similar health or human activity sensing tasks (and not age and sex classification task). Unfortunately, the generative tasks including random imputation, temporal interpolation, sensor imputation, and temporal extrapolation for forecasting are the same tasks that you have used for pre-training. I am curious to see more independent downstream tasks. Regarding concern (3), just because a previous paper included age and sex classification task, does it make it ethical? Can you please confirm that you received IRB or ethical review board permission to develop predictive models for biological sex and age from wearable data with your proposed foundation model? In the clinical context, would the foundation model conform to HIPAA regulations?
> > > >
> > > > I recognize that concern (2) is difficult to address and it was not your initial goal for this paper. I am ok to ignore it but can you please talk about this point in your discussion?

---

> > > > > ### Author Response · Authors · 2024-11-27
> > > > > **Response To Concerns (1/2)**
> > > > >
> > > > > Thanks for working through this process with us. We’ve responded to your concerns below.
> > > > > # 1) Generalizability of the Proposed Model
> > > > > We agree that additional diverse tasks would further highlight the generalizability of the model, and we are glad that you feel the newly added mood task is a step in that direction. We are, regrettably, limited by our dataset, which hosts activity, mood, and demographic labels. We have added acknowledgement of this to **Section 6 Limitations and Future Work**. We would also like to clarify, that though the generative tasks (of temporal interpolation, temporal extrapolation, and sensor imputation) are conceptually similar to our generative pretraining task of random interpolation (reconstruction of randomly masked patches), these downstream tasks differ in that they gauge the model‘s ability to infer highly structured missing data and are qualitatively different from the random masking pretraining  task.
> > > > >
> > > > > # 2) Generalizing to Other Devices and Sensor Arrays
> > > > > Thank you for understanding  the practical constraints of our work. We have added acknowledgement of this in **Section 6 Limitations and Future Work** and have added discussion of this in **Appendix D.**

---

> > > > > > ### Author Response · Authors · 2024-11-27
> > > > > > **Response To Concerns (2/2)**
> > > > > >
> > > > > > # 3) Regarding  Ethical Considerations of Our Methods
> > > > > > **We would like to make it clear that we have completed a review through an independent Institutional Review Board (IRB). This research was approved and the data used in the research were considered deidentified. The research participants provided informed consent as documented in the submitted IRB protocol. We are happy to provide any additional detail about the IRB or consent.**
> > > > > >
> > > > > > Regarding the ethics of predicting age/gender, we understand the motivation behind the concern. First, we should clarify that **age (in years and not finer-grained), and gender/sex are not HIPAA identifiers** [1] and are often included in deidentified datasets. Second, prediction of these variables differs substantially from attempts towards patient re-identification (which we agree can have ethical implications). Third, predicting demographic variables can help with detecting patient mis-identification errors [2, 3], and is commonly done in understanding chronological vs. biological aging [4, 5]. Lastly, whether foundation models should contain information about demographics is a nuanced question; this is further discussed in prior work [6].
> > > > > >
> > > > > > We would like to once again emphasize that **our methods were not specifically designed to capture demographic  information like age and biological sex**. Such attributes are highly correlated to a person’s physiology. For example, it is well studied that the performance of an individual's cardiovascular system changes dramatically as they age. It is for this reason that cardiovascular disease is the most frequent cause of death for only older individuals (65+) [7]. Thus, it follows that methods trained on wearable  PPG (pulse data) would exhibit some understanding  of a person’s demographic information. Similarly, it is well known that a person’s maximal heart rate is strongly correlated with age, with studies finding correlations as strong as **r = -0.9** with simple linear models [8]. Thus, logically, more complex systems, which capture physiological features, would naturally, to some degree, capture demographic  features correlated to the underlying physiology. Prior works [9, 10] investigate the ability of models’ to predict demographic  features. In doing so they highlight both the associated **risks of these capabilities** as well as the **importance of conducting such investigations**. We find their discussion of these ethical considerations insightful and have accordingly added discussion around this point and referred readers to these works in **Appendix D.**
> > > > > >
> > > > > > Health is a sensitive topic, so we appreciate your commitment to ensuring that our work follows ethical guidelines. As these experiments were **specifically asked for by other reviewers** we do not find it immediately obvious how to proceed next. As such, in addition to these tasks we have added robust discussion around the safety and ethical considerations needed when working with such health foundation models. These discussions can be found in **Appendix D: Additional Discussions, Limitations, and Future Work** and in **Appendix H: Broader Impact.**
> > > > > >
> > > > > > ## References
> > > > > > [1] https://www.dhcs.ca.gov/dataandstats/data/Pages/ListofHIPAAIdentifiers.aspx
> > > > > > [2] Ueda, Yasuyuki, and Junji Morishita. "Patient identification based on deep metric learning for preventing human errors in follow-up X-ray examinations." Journal of Digital Imaging 36.5 (2023): 1941-1953.
> > > > > > [3] Kim, Kiduk, et al. "Screening Patient Misidentification Errors Using a Deep Learning Model of Chest Radiography: A Seven Reader Study." Journal of Imaging Informatics in Medicine (2024): 1-9.
> > > > > > [4] Raghu, Vineet K., et al. "Deep learning to estimate biological age from chest radiographs." Cardiovascular Imaging 14.11 (2021): 2226-2236.
> > > > > > [5] Mitsuyama, Yasuhito, et al. "Chest radiography as a biomarker of ageing: artificial intelligence-based, multi-institutional model development and validation in Japan." The Lancet Healthy Longevity 4.9 (2023): e478-e486.
> > > > > > [6] Weng, Wei-Hung, et al. "An intentional approach to managing bias in general purpose embedding models." The Lancet Digital Health 6.2 (2024): e126-e130.
> > > > > > [7] Wei, Jeanne Y. "Age and the cardiovascular system." New England Journal of Medicine 327.24 (1992): 1735-1739.
> > > > > > [8] Tanaka, Hirofumi, Kevin D. Monahan, and Douglas R. Seals. "Age-predicted maximal heart rate revisited." Journal of the american college of cardiology 37.1 (2001): 153-156.
> > > > > > [9] Gichoya, Judy Wawira, et al. "AI recognition of patient race in medical imaging: a modelling study." The Lancet Digital Health 4.6 (2022): e406-e414.
> > > > > > [10] Glocker, Ben, et al. "Algorithmic encoding of protected characteristics in chest X-ray disease detection models." EBioMedicine 89 (2023).

---

> > > > > > > ### Comment · Reviewer_ZN2N · 2024-11-28
> > > > > > >
> > > > > > > Thanks for confirming that you have IRB approvals for all the modeling and analyses that you have performed including the age and biological sex classification with wearable data.

---

### Author Response · Authors · 2024-11-21
**Overview and TLDR**

We’d like to thank all of the reviewers for taking the time and effort to provide such high quality feedback. We strongly believe that this process has helped strengthen the quality and clarity of our work. We have responded to each of you individually and would like to highlight the changes we’ve made throughout this rebuttal process (these changes are reflected as blue text in the newly uploaded paper file).

**1. Additional Discriminative Downstream Tasks.** As requested by many reviewers, we have added additional tasks including sex, and age classification to better show the generalizability of our model. The description of these tasks and the table results can be found in **Appendix Section C.7.**

**2. Addition of a More Sophisticated Generative Baseline.** As requested by multiple reviewers, and based on the suggestion of reviewer 58g1, we add MICE (Multivariate Imputation by Chained Equations), and compare its performance against LSM and our other baselines in **Table 3.**

**3. Addition of Classical ML Baselines.** Based on the recommendation of reviewer Y2nv, we add Random Forest and Logistic Regression Classifier baselines to our discriminative tasks. The performance of these methods contrasted against LSM and existing baselines can be found in **Table 3** of the main paper and **Table 14** of the appendix.

**4. Quantitative Analysis of Embeddings.** Based on the recommendation of reviewer Y2nv, we add numerics to augment the tSNE embedding plots we present in the appendix. These can be found in **Table 12.**

**5. Clarification of Datasets, Methods, and Assumptions.** Several reviewers had questions regarding statistics of our dataset and its distributions, our sampling techniques, the choice of 1-minute granular features, our choice of 5-hour windows, our handling of imputed data and more. To this end we have clarified information regarding our dataset and its distributions (**Sections 3.1, 3.2, 4.2, 6**, **Appendix C.7, G.2, G.3**). We have added details regarding our modeling methods (**Section 5.1, Appendix E.3**), our choice of 5 our windows and minutely features (**Section 3.1, 6**). And we have stated additional assumptions (**Section 6, Appendix D, G.4**). We have additionally gone through and improved the writing of our feature descriptions (**Section 3.1** and **Appendix Table 18**).

**6. Added Discussion.** Motivated by several reviewers we add additional discussion regarding the practicality of scaling, the challenges, and moving such large models to the edge. This can be found in **Section 6** and **Appendix D.1**

**7. Relevant Related Work** Several reviewers have pointed to relevant related work that help better contextualize our work. We add these references in **Sections 5.1**, and **Appendix B.**

---

> ### Author Response · Authors · 2024-11-25
> **Added Task (Mood)**
>
> Hi all, thanks again for helping us improve this work! In an effort to expand the diversity of our downstream tasks (as requested by several reviewers) we have added the discriminative task of subject dependent mood recognition. Details and results may be found in **Appendix C.7 and Table 14.**

---

### Meta-Review · Area_Chair_sg9c · 2024-12-21

**Metareview:**

This paper proposes a pretrained foundation model based on a large-scale wearable sensor data from 165,000 individuals, collected over than 40,000 hours, consisting of multiple sensor modalities, i.e. accelerometer, PPG, EDA, skin temperature, and altimeter sensor. This is the largest pretrained foundation model on sensor data to date. As such, the authors provided new insights into the scaling laws and training strategies applicable to wearable sensor data, including sample scaling and subject scaling.
Initially, there were a shared concern by the reviewers that the generality of the foundation model is an overclaim, given the limited downstream tasks, only on imputation, prediction, and discriminative tasks. There are also concerns that the data is not openly shared.

During the rebuttal phase, the authors have also added new experiments on new downstream tasks, including mood classification, age classification, and sex classification -- increasing the diversity of the tasks, strengthening the claim of generality. The authors have also added more details on the method and experiments, and have better contextualised the work among other similar works in the recent years, and added discussions on limitations.

The insights into model saturation is also very interesting.

I believe this paper should be accepted, given the extensive and comprehensive effort, and the strong rebuttal. The contributions of this paper will guide other researchers in the same field, especially those who may not have the same access to the large scale dataset or compute. This paper will also lead to interesting discussion on foundation models training beyond text.

**Additional Comments On Reviewer Discussion:**

No further discussion. However, during the rebuttal phase, most reviewers increased their score.

---

### Decision · Program_Chairs · 2025-01-22

Accept (Poster)